# A novel L1CAM isoform with angiogenic activity generated by NOVA2-mediated alternative splicing

Francesca Angiolini[1†§], Elisa Belloni[2†], Marco Giordano[1†], Matteo Campioni[3], Federico Forneris[3], Maria Paola Paronetto [4], Michela Lupia[1], Chiara Brandas[2], Davide Pradella[2,5], Anna Di Matteo[2], Costanza Giampietro[6#], Giovanna Jodice[7], Chiara Luise[7], Giovanni Bertalot[7], Stefano Freddi[7], Matteo Malinverno[6], Manuel Irimia[8,9,10], Jon D Moulton[11], James Summerton[11], Antonella Chiapparino[3], Carmen Ghilardi[12], Raffaella Giavazzi[12], Daniel Nyqvist[13], Davide Gabellini[14], Elisabetta Dejana[6,15], Ugo Cavallaro[1‡*], Claudia Ghigna[2‡*]

[1]Unit of Gynecological Oncology Research, Program of Gynecological Oncology, IEO, European Institute of Oncology IRCCS, Milan, Italy; [2]Istituto di Genetica Molecolare, Consiglio Nazionale delle Ricerche, Pavia, Italy; [3]The Armenise-Harvard Laboratory of Structural Biology, Department of Biology and Biotechnology, University of Pavia, Pavia, Italy; [4]Department of Movement, Human and Health Sciences, Università degli Studi di Roma "Foro Italico", Rome, Italy; [5]Università degli Studi di Pavia, Pavia, Italy; [6]FIRC Institute of Molecular Oncology, Milan, Italy; [7]Molecular Medicine Program, IEO, European Institute of Oncology IRCCS, Milan, Italy; [8]Centre for Genomic Regulation, The Barcelona Institute of Science and Technology, Barcelona, Spain; [9]Universitat Pompeu Fabra, Barcelona, Spain; [10]Institució Catalana de Recerca i Estudis Avançats, Barcelona, Spain; [11]Gene Tools LLC, Philomath, United States; [12]Laboratory of Biology and Treatment of Metastasis, IRCCS-Istituto di Ricerche Farmacologiche Mario Negri, Milan, Italy; [13]Division of Vascular Biology, Department of Medical Biochemistry and Biophysics, Karolinska Institutet, Stockholm, Sweden; [14]Division of Genetics and Cell Biology, IRCCS San Raffaele Scientific Institute, Milan, Italy; [15]Rudbeck Laboratory and Science for Life Laboratory, Department of Immunology, Genetics and Pathology, Uppsala University, Uppsala, Sweden

*For correspondence:
ugo.cavallaro@ieo.it (UC);
arneri@igm.cnr.it (CG)

†These authors contributed equally to this work
‡These authors also contributed equally to this work

Present address: §GSK Vaccines Srl, Siena, Italy; #Laboratory of Thermodynamics in Emerging Technologies, Department of Mechanical and Process Engineering, ETH Zurich, Zurich, Switzerland

**Abstract** The biological players involved in angiogenesis are only partially defined. Here, we report that endothelial cells (ECs) express a novel isoform of the cell-surface adhesion molecule L1CAM, termed L1-ΔTM. The splicing factor NOVA2, which binds directly to *L1CAM* pre-mRNA, is necessary and sufficient for the skipping of L1CAM transmembrane domain in ECs, leading to the release of soluble L1-ΔTM. The latter exerts high angiogenic function through both autocrine and paracrine activities. Mechanistically, L1-ΔTM-induced angiogenesis requires fibroblast growth factor receptor-1 signaling, implying a crosstalk between the two molecules. NOVA2 and L1-ΔTM are overexpressed in the vasculature of ovarian cancer, where L1-ΔTM levels correlate with tumor vascularization, supporting the involvement of NOVA2-mediated L1-ΔTM production in tumor angiogenesis. Finally, high NOVA2 expression is associated with poor outcome in ovarian cancer patients. Our results point to L1-ΔTM as a novel, EC-derived angiogenic factor which may represent a target for innovative antiangiogenic therapies.
DOI: https://doi.org/10.7554/eLife.44305.001

**eLife digest** Growing tumors stimulate the formation of new blood vessels to supply the oxygen and nutrients the cancerous cells need to stay alive. Stopping tumors from forming the blood vessels could therefore help us to treat cancer. To do so, we need to understand how different proteins control when and how blood vessels develop.

Cells make proteins by first 'transcribing' genes to form RNA molecules. In many cases, the RNA then goes through a process called alternative splicing. Proteins known as splicing factors cut out different segments of the RNA molecule and stick together the remaining segments to form templates for protein production. This enables a single gene to produce many different variants of a protein.

Angiolini, Belloni, Giordano et al. have now studied mouse and human versions of the cells that line the blood vessels grown by tumors. This revealed that a splicing factor called NOVA2 targets a protein called L1CAM, which is normally responsible for gluing adjacent cells together. Angiolini et al. found that NOVA2 splices L1CAM into a form not seen before. Instead of remaining anchored to cell surfaces, the newly identified form of L1CAM is released into the blood circulation, where it stimulates new blood vessels to grow.

Samples taken from the blood vessels of human ovarian tumors showed high levels of both NOVA2 and the modified form of L1CAM, while blood vessels in healthy tissue contain no, or very low levels of both proteins. Therefore, if the new form of L1CAM can be detected in the blood, it could be used to help cancer diagnosis, and to indicate which patients would benefit from treatments that restrict the growth of blood vessels in tumors. Further work is now needed to explore these possibilities.

DOI: https://doi.org/10.7554/eLife.44305.002

## Introduction

L1CAM, also known as CD171 or L1, is a cell adhesion molecule encoded by the gene *L1CAM*. It is a cell surface glycoprotein with an extracellular portion that comprises six Ig-like domains, five fibronectin type III repeats, a short transmembrane (TM) domain, and a conserved cytoplasmic tail. A disintegrin and metalloproteinase (ADAM)10-mediated cleavage at membrane proximal site induces shedding of the L1CAM ectodomain (*Mechtersheimer et al., 2001*), while intramembrane processing mediated by γ-secretase generates a cytosolic domain which can translocate to the nucleus and modulate gene expression (*Maretzky et al., 2005*). L1CAM was initially identified in the nervous system and characterized for its important function in neural development and plasticity (*Maness and Schachner, 2007*). Further studies then showed that L1CAM is not restricted to the nervous system and, in particular, we reported its expression in the tumor vasculature of several cancer types, while no or very low L1CAM expression is detectable in normal vessels (*Maddaluno et al., 2009*; *Magrini et al., 2014*). L1CAM orchestrates different endothelial cell (EC) functions within tumor-associated vessels, such as permeability, pericyte coverage and polarity. Since these cellular processes influence tumor angiogenesis, cancer growth and metastasis, L1CAM has emerged as a potential target for tumor vascular-specific therapies (*Magrini et al., 2014*).

L1CAM occurs mainly in two alternatively spliced isoforms: while neurons typically express the full-length variant of L1CAM, non-neural cell types produce a shorter isoform which lacks exon 2 and exon 27. Exon 2 is involved in the interaction with other neuronal proteins, and exon 27 facilitates endocytosis of L1CAM (*Schäfer and Altevogt, 2010*).

Alternative splicing (AS) produces different mature transcripts (mRNAs) from a single primary pre-mRNA. AS decisions are modulated by a number of *cis*-acting motifs and splicing regulatory factors (SRFs) that function in a coordinate manner to promote or inhibit the inclusion of specific exons into the mRNA (*Fu and Ares, 2014*; *Nilsen and Graveley, 2010*). More than 90% of human protein-coding genes undergo AS (*Pan et al., 2008*; *Wang et al., 2008*) giving rise to different protein isoforms with distinct structural and functional properties. Hence, AS represents an important mechanism to expand the coding potential of the human genome, thus contributing to generate the cellular

complexity of different tissue types and to support key functional properties (*Chen and Manley, 2009*; *Baralle and Giudice, 2017*). Notably, several findings highlighted a direct role of AS in promoting cancer progression (*Anczuków and Krainer, 2016*; *Biamonti et al., 2014*; *Pradella et al., 2017*). In particular, it has been shown that mutations or altered expression of specific SRFs allow neoplastic cells to generate cancer-specific AS isoforms involved in tumor establishment, progression and resistance to therapeutic treatments (*Bonomi et al., 2013a*; *Anczuków and Krainer, 2016*; *Biamonti et al., 2014*; *Oltean and Bates, 2014*). These 'oncogenic AS switches' can be used to stratify patients according to tumor stage (*Stricker et al., 2017*; *Inoue and Fry, 2015*), while their targeting represents a promising approach to improve the efficacy of anti-cancer treatments (*Bonomi et al., 2013a*; *Agrawal et al., 2018*; *Anczuków and Krainer, 2016*). However, in contrast to the established role of AS in tumor cells, it remains unclear whether this process is also relevant in tumor microenvironment and, in particular, in cancer vasculature. In fact, AS events specifically occurring in tumor-associated ECs have been described (*Neri and Bicknell, 2005*) and proposed as potential targets for antiangiogenic therapies (*Steiner and Neri, 2011*). However, how such AS events impact on the pathophysiology of tumor vasculature remains elusive.

Recently, we described the SRF Neuro-Oncological Ventral Antigen 2 (NOVA2) as a prominent regulator of AS during vascular development (*Giampietro et al., 2015*). NOVA2 was initially identified in neural cells where it controls AS of several genes involved in various neural developmental processes by binding to clusters of YCAY (Y = C/U) repeats within its pre-mRNA targets (*Licatalosi et al., 2008*; *Ule et al., 2003*; *Zhang et al., 2010*; *Leggere et al., 2016*; *Saito et al., 2016*). Our study revealed that NOVA2 is also expressed in vascular endothelium and is regulated during angiogenesis (*Giampietro et al., 2015*). NOVA2 controls at the post-transcriptional level the establishment of EC polarity, a process that is essential for vascular lumen formation and, hence, for angiogenesis (*Iruela-Arispe and Davis, 2009*). Accordingly, NOVA2 ablation causes defects in vascular lumen formation in vivo (*Giampietro et al., 2015*).

Here, we report a novel isoform of L1CAM expressed in ECs as the result of a NOVA2-induced AS event that removes the exon encoding the transmembrane domain of the protein. This gives rise to a soluble L1CAM variant, referred to as L1-ΔTM, that is released by ECs and is able to stimulate angiogenesis via autocrine/paracrine mechanisms. NOVA2 and L1-ΔTM are overexpressed in the vasculature of ovarian cancer and correlate with poor outcome and tumor vascularization, respectively. Our findings, therefore, implicate the novel NOVA2/L1-ΔTM axis in EC pathophysiology and in ovarian cancer aggressiveness.

## Results

### Alternative splicing of *L1CAM* in endothelium

We have recently reported the novel function of L1CAM in vascular endothelium (*Magrini et al., 2014*). Since AS is known to influence the biological activities of cell-surface adhesion molecules (*Wang et al., 2005*), it is possible that AS of *L1CAM* accounts for, or at least contributes to, its peculiar role in ECs. A bioinformatics analysis with the ExonMine program (http://www.imm.fm.ul.pt/exonmine/) (*Mollet et al., 2010*) identified a human expressed sequence tag (EST) in which the *L1CAM* exon 25 (a 135-nucleotide cassette exon) is excluded from the mature mRNA (*Figure 1A*). We then analyzed several normal human tissues and human ECs for the AS of human *L1CAM* exon 25 by RT–PCR (*Figure 1B*). In addition, we also investigated the AS of this exon in the mouse. In the murine gene, this exon is annotated as exon 26 by UCSC and Ensembl, due to the presence of an additional non-coding exon upstream of exon 1 (i.e., the one containing the ATG codon). Nevertheless, based on its high homology to the human *L1CAM* exon 25 (89% identity), we refer to it as exon 25 also in mouse *L1cam*. The AS of this exon was examined in normal mouse tissues, mouse EC lines and freshly purified murine ECs. As shown in *Figure 1B and C*, in both human and mouse samples the skipping of exon 25 mainly occurred in ECs. Overall, these data suggest that ECs express a novel alternatively spliced isoform of *L1CAM* devoid of exon 25.

### Alternative splicing generates a novel soluble form of L1CAM

Skipping of exon 25 results in an in-frame deletion of a 44-amino acid sequence (45 in mouse) that encompasses the entire transmembrane (TM) domain of L1CAM (*Figure 2A*). This suggests that AS

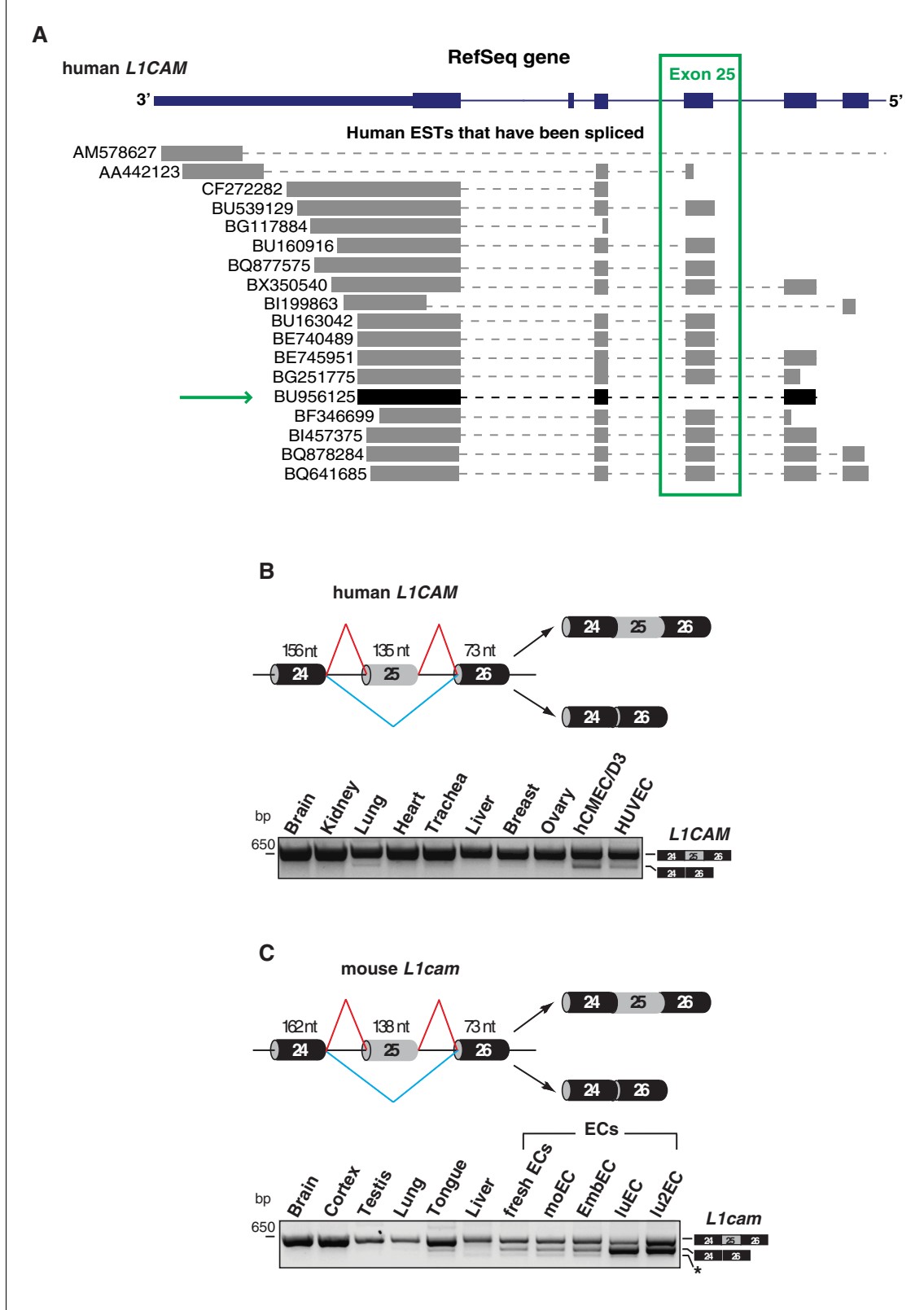

**Figure 1.** Alternative splicing of *L1CAM* exon 25. (**A**) 3' region of human *L1CAM* gene and AS variants that are present as ESTs (data from UCSC Genome Browser). The green box indicates exon 25. The green arrow shows an EST with the skipping of exon 25. The annotation of human *L1CAM* exon 25 refers to RefSeq transcript NM_00425 and is consistent with the previous literature (*Mikulak et al., 2012*). (**B**) Upper panel: schematic diagram of the human *L1CAM* genomic region containing the AS exon 25 (grey box). Black boxes = constitutive exons; thin lines = introns. Red and blue lines

*Figure 1 continued on next page*

## eLIFE Research article

Cancer Biology | Cell Biology

*Figure 1 continued*

indicate the two possible AS reactions, and the two resulting isoforms are shown on the right. Lower panel: RT-PCR analysis of AS of the human *L1CAM* exon 25 in different human tissues and in two EC lines (hCMEC/D3 and HUVEC). (C) Scheme of AS events of the mouse *L1cam* exon 25 and RT-PCR analysis in mouse tissues, ECs freshly purified from mouse lung and mouse EC lines (moEC, EmbEC, luEC and lu2EC). Asterisk indicates an additional band (more evident in moEC and EmbEC) corresponding to a novel transcript deleted of exons 25, 26 and 27.

DOI: https://doi.org/10.7554/eLife.44305.003

The following figure supplement is available for figure 1:

**Figure supplement 1.** *L1cam* splicing in mouse and human ECs.

DOI: https://doi.org/10.7554/eLife.44305.004

of exon 25 could affect L1CAM localization and, hence, its activity. To test this hypothesis, we selected an immortalized endothelial cell line of murine origin, moEC (*Lampugnani et al., 2002*; *Taddei et al., 2008*), because it expresses no or very little endogenous L1CAM (*Figure 2B*) and, therefore, is amenable to gain-of-function studies. As shown in *Figure 1—figure supplement 1*, ECs express the non-neural isoform of *L1CAM* mRNA, which lacks exons 2 and 27 (*Schäfer and Altevogt, 2010*). Thus, exon 25 was deleted from non-neural mouse *L1cam* in order to obtain L1-ΔTM. MoEC were stably transduced with either full-length, non-neural L1CAM (L1-FL) or with L1-ΔTM. Immunoblotting of total extracts from L1-FL-expressing cells showed the expected doublet pattern (*Figure 2B*), with the upper band representing the mature, fully glycosylated cell-surface form, and the lower band corresponding to the precursor form (*Moulding et al., 2000*; *Zisch et al., 1997*). In contrast, L1-ΔTM-expressing moEC showed a single band that migrated slightly faster than the lower band of L1-FL (*Figure 2B*), consistent with the deletion of exon 25. The lack of the TM domain might affect the subcellular localization of L1CAM. To address this question, we performed immunofluorescence staining for L1CAM on moEC expressing the two isoforms. Only L1-FL was prominently exposed on the cell surface of moEC, while cells expressing L1-ΔTM exhibited mainly a cytoplasmic localization (*Figure 2C*). Based on the deletion of the TM domain and on the absence of membrane staining, we hypothesized that L1-ΔTM is released into the extracellular space and represents a novel soluble form of the protein. Indeed, conditioned medium (CM) from L1-ΔTM-expressing moEC contained high amounts of L1-ΔTM (*Figure 2D*). Of note, the protein in the CM retained the cytoplasmic tail (*Figure 2—figure supplement 1*). This confirmed the release of the entire molecule as opposed to the proteolytic cleavage of full-length L1CAM, which results in the shedding of its extracellular portion (*Figure 2D*; *Mechtersheimer et al., 2001*). Both the cytoplasmic localization of L1-ΔTM and its release into the CM were also confirmed in luEC, another murine EC line (*Figure 2—figure supplement 1*).

Interestingly, L1-ΔTM was detected in the culture medium as a single band that migrated slower than the protein found in the cell lysate (*Figure 2D*). We hypothesized that the different size was accounted for by the glycosylation state of the protein. This was confirmed by the forced expression of L1-ΔTM in N-acetylglucosaminyltransferase I-deficient HEK293 cells [GnTI(-)] (*Reeves et al., 2002*). The latter, indeed, released a form of L1-ΔTM that exhibited a lower molecular weight as compared to wild-type HEK293 cells (*Figure 2—figure supplement 1*). These findings support the notion that L1-ΔTM is released in its mature, fully glycosylated form.

In order to confirm the release of endothelial L1-ΔTM in an endogenous system, we employed lu2EC, an immortalized mouse endothelial cell line that expresses relatively high levels of *L1cam* (*Figure 2—figure supplement 1*). These cells also express endogenous *L1-ΔTM* (*Figure 2—figure supplement 1*). When the lu2EC-derived CM was immunoblotted with the antibody against the cytoplasmic tail of L1CAM, we found high levels of L1-ΔTM (*Figure 2—figure supplement 1*), confirming its release into the extracellular space. To compare our findings in ECs with a non-endothelial cell type, we used the mouse melanoma cell line B16, which also expresses high levels of endogenous L1CAM (*Linnemann and Bock, 1986*; *Magrini et al., 2014*), but does not express *L1-ΔTM* (*Figure 2—figure supplement 1*). While we could detect high levels of cell-associated L1CAM, no C-terminus-containing L1CAM was detected in the CM of B16 cells (*Figure 2—figure supplement 1*), supporting the hypothesis that the AS of L1CAM results in the release of L1-ΔTM in ECs.

Collectively, our results indicate that skipping of *L1CAM* exon 25 generates a novel isoform of the protein that is released in the extracellular compartment.

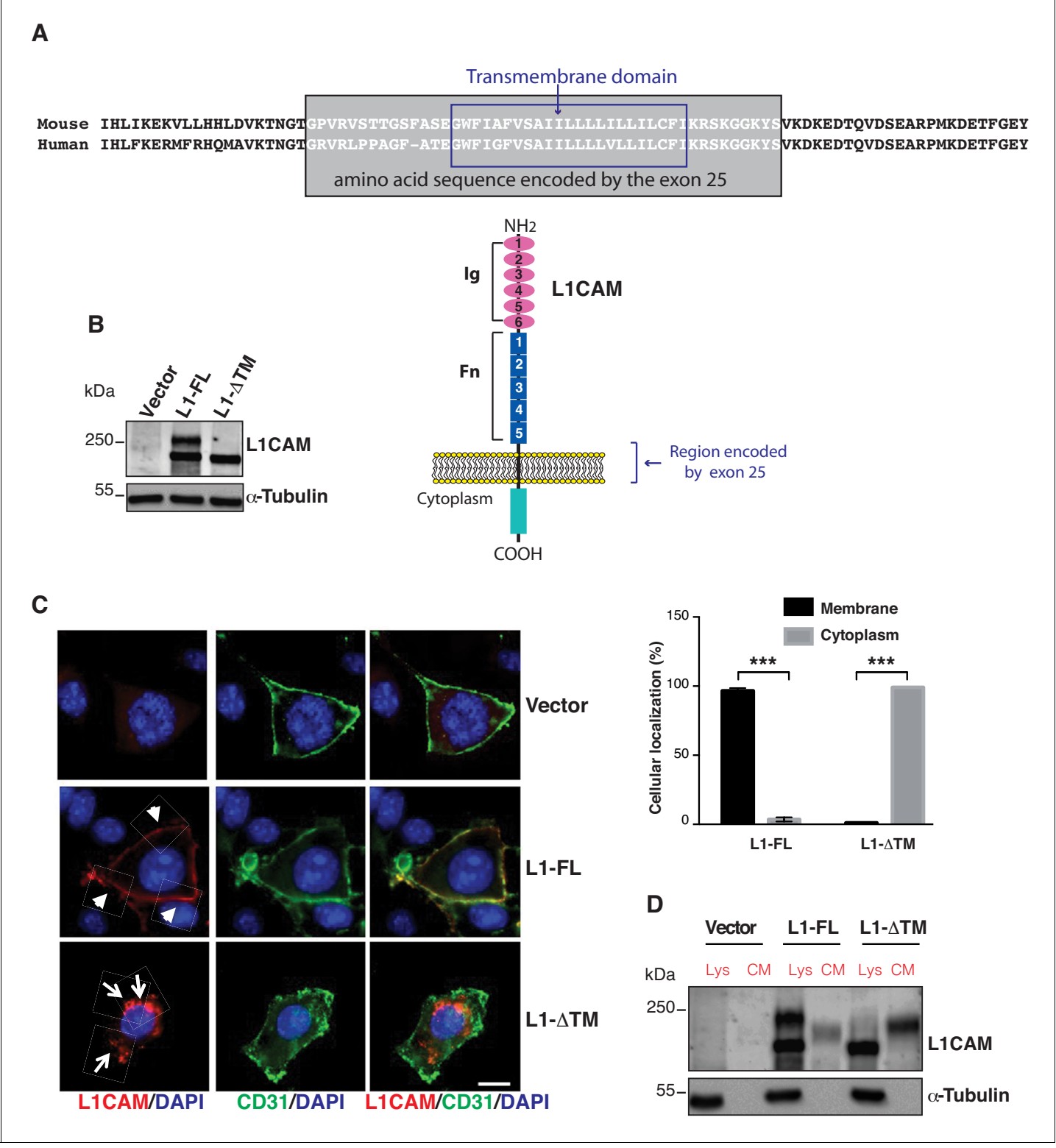

**Figure 2.** Expression, cell surface localization and release of L1CAM isoforms. (**A**) The amino acid sequence of the mouse and human L1CAM region across the membrane. The transmembrane domain (blue rectangle), with 91% identity between mouse and human, and the sequence encoded by exons 25 (grey rectangle) are indicated. Bottom: schematic structure of L1CAM, showing the six Ig domains (Ig) and the five FN type-III repeats (Fn) in the extracellular portion. (**B**) Immunoblotting for L1CAM on lysates from moEC stably over-expressing the L1CAM isoforms (L1-FL or L1-ΔTM) or the empty vector (Vector). Immunoblotting for α-Tubulin served as loading control. (**C**) Representative images from the immunofluorescence analysis of

*Figure 2 continued on next page*

*Figure 2 continued*

L1CAM (red) and the endothelial cell surface marker CD31 (green) on moEC overexpressing either L1-FL or L1-ΔTM (confocal sections, z axis; scale bar 10 µm). Arrowheads show L1-FL localization at the cell surface, while arrows show the cytosolic localization of L1-ΔTM. The graph (right panel) shows the quantitation of the cellular localization of the two L1CAM isoforms. Values represent means ±SD from five different fields in each condition. Comparisons between experimental groups were done with two-sided Student's t-test; ***p<0.001. (D) Immunoblotting for L1CAM on lysates (Lys) and conditioned media (CM) from moEC stably overexpressing either L1-FL or L1-ΔTM. Equal amounts of protein extracts and volumes of CM derived from equal numbers of producing cells (see Materials and methods) were analysed.

DOI: https://doi.org/10.7554/eLife.44305.005

The following figure supplement is available for figure 2:

**Figure supplement 1.** Characterization of L1-ΔTM isoform.

DOI: https://doi.org/10.7554/eLife.44305.006

## L1-ΔTM regulates endothelial cell function

To investigate the biological role of the L1-ΔTM isoform in ECs, we focused on their ability to form capillary-like tubes in three-dimensional matrices, which reflects their angiogenic potential (*Di Blasio et al., 2014*). Therefore, we assayed control and L1-ΔTM-expressing ECs for tube formation on Matrigel. As shown in *Figure 3—figure supplement 1*, *Figure 3A* and *Video 1*, L1-ΔTM enhanced significantly the tube forming ability of moEC, thus suggesting that it is endowed with angiogenic properties. The direct role of L1-ΔTM in moEC tube formation was probed with 324, a L1CAM-neutralizing antibody (*Appel et al., 1993*; *Di Sciullo et al., 1998*). As shown in *Figure 3B*, the antibody 324, but not a control antibody, abolished the tube-forming potential of moEC. The results of this proof-of-concept experiment also support the neutralization of vascular L1-ΔTM as a potential strategy to interfere with the angiogenic process.

Based on the substantial release of L1-ΔTM into the extracellular space, we asked whether the molecule could also exert its biological function as a soluble factor in a paracrine fashion. To address this question, parental moEC were subjected to tube formation assays in the presence of the CM from moEC expressing either L1-ΔTM or the control vector. ECs exposed to the CM from L1-ΔTM-expressing cells exhibited higher tube-forming activity than those exposed to control medium (*Figure 3C*) or to the CM from L1-FL-expressing cells (not shown). Similar results were obtained by using CM from luEC expressing either L1-ΔTM or the control vector (*Figure 3—figure supplement 1*). To further verify the angiogenic activity of soluble L1-ΔTM, we treated parental moEC with a purified, recombinant version of the protein produced in mammalian cells (*Figure 3—figure supplement 1*). Indeed, recombinant soluble L1-ΔTM induced moEC tube formation in a dose-dependent manner (*Figure 3D*), thus confirming its ability to stimulate EC remodeling and morphogenesis. In order to validate our results in an EC model with endogenous L1-ΔTM, we treated lu2EC with a morpholino oligonucleotide that selectively prevents the inclusion of *L1cam* exon 25 (*Figure 3E*). As shown in *Figure 3F*, this resulted in increased expression and extracellular release of endogenous L1-ΔTM. Importantly, lu2EC exposed to the CM from morpholino-treated cells exhibited higher tube-forming activity than those exposed to control CM, thus confirming the functionality of endogenous L1-ΔTM (*Figure 3F*).

Finally, we aimed at validating our findings in an in vivo assay of angiogenesis. Mice underwent subcutaneous implantation of Matrigel plugs containing CM from either L1-ΔTM- or L1-FL-expressing ECs or from control cells. Neovascularization was markedly induced by L1-ΔTM-containing CM, while a weaker effect was observed with the CM from L1-FL-expressing cells (*Figure 3G*). This strongly supports the angiogenic function of L1-ΔTM.

## FGFR1 signaling is required for L1-ΔTM-induced tube formation

Previous studies implicated fibroblast growth factor receptor (FGFR) signaling as an effector of L1CAM in different cellular contexts (*Díaz-Balzac et al., 2015*; *Kulahin et al., 2008*; *Mohanan et al., 2013*; *Williams et al., 1994*; *Zecchini et al., 2008*). However, the L1CAM/FGFR interplay in ECs has not been investigated. Given the well-characterized role of FGFR function in vascular biology and angiogenesis (*Ronca et al., 2015*), we hypothesized that the pro-angiogenic effect of L1-ΔTM was mediated by FGFR. Among the four FGFR family members, moEC express only *FGFR1* (data not shown) (*Giampietro et al., 2012*), as previously reported for other EC types (*Giacomini et al., 2016*;

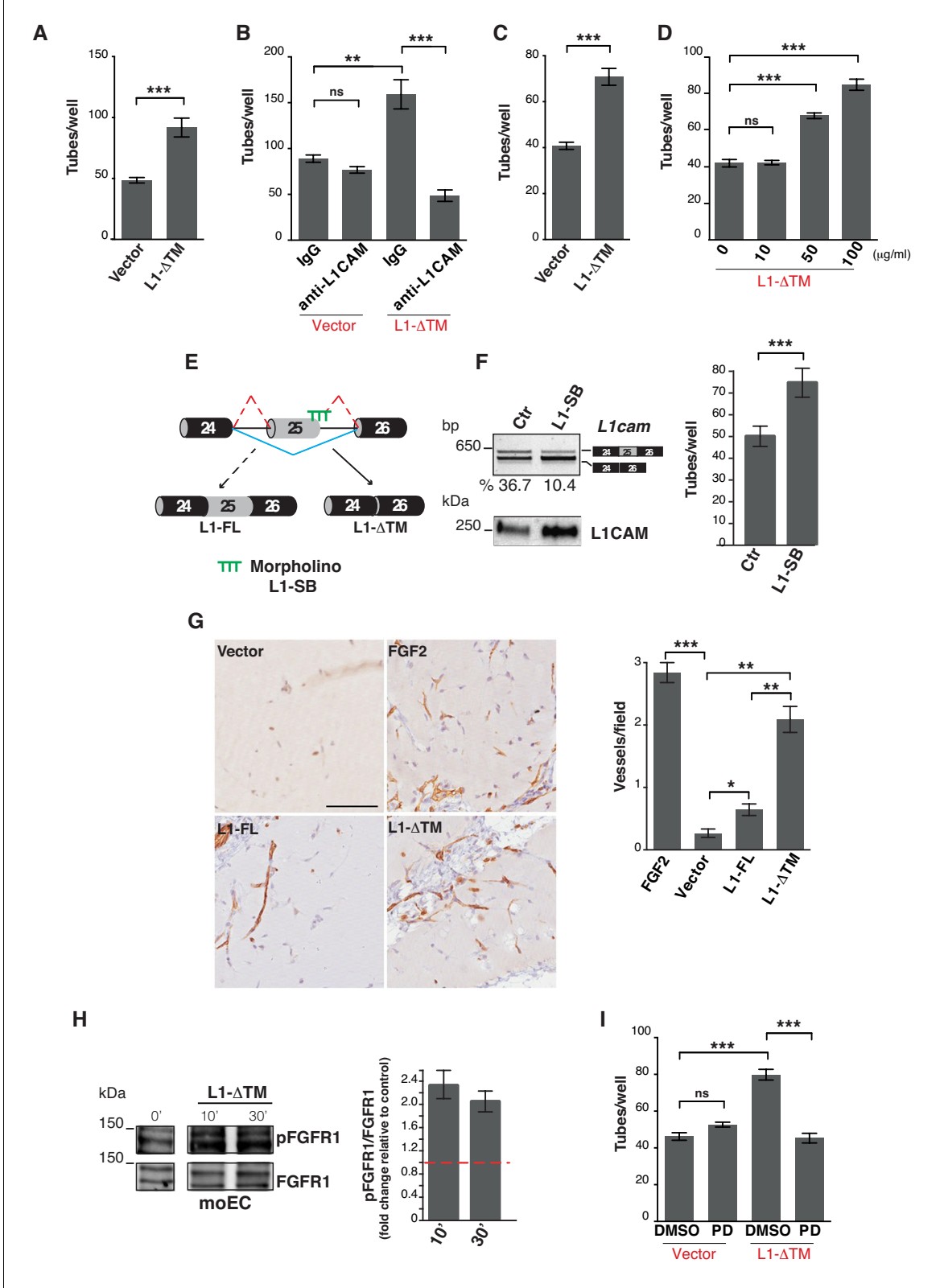

**Figure 3.** L1-ΔTM stimulates angiogenesis in an autocrine/paracrine fashion. (**A**) Quantitation of tube formation by transduced moEC. (**B**) Transduced moEC were subjected to tube formation assays in the presence of either anti-L1CAM clone 324 or a control, irrelevant antibody (IgG). (**C**) Quantitation of tube formation by parental moEC treated with CM from moEC transduced either with the empty vector (Vector) or with L1-ΔTM. (**D**) Quantitation of tube formation assays on moEC untreated or treated with increasing concentrations of recombinant L1-ΔTM. (**E**) Schematic illustration of the

*Figure 3 continued on next page*

*Figure 3 continued*

mechanism of action of the morpholino oligonucleotide (**L1–SB**), which binds to the exon 25/intron 25 junction of *L1cam*, thus preventing the recruitment of the spliceosome and, hence, impairing the inclusion of exon 25. (**F**) lu2EC transfected with either an irrelevant morpholino (Ctr) or with L1-SB were analyzed by RT-PCR for the AS of *L1cam* exon 25 (left, top panel), whereas CM from the same cells were analyzed in immunoblotting with the L1CAM antibody (left, bottom panel). Parental moEC were subjected to tube formation assays in the presence of CM from Ctr- or L1-SB-transfected lu2EC (right panel). (**G**) Representative images and quantitation of vessel density in matrigel plugs pre-mixed with the CM from ECs transduced with either the empty vector (Vector), L1-FL or L1-ΔTM, and then implanted subcutaneously into C57Bl/6 mice (n = 3 mice/group). Matrigel plugs containing FGF2 served as positive control. Scale bar, 100 μm. Right panel: CD31[+] vessels were counted in five different fields. (**H**) Left panels: immunoblots for phospho-FGFR1 (pFGFR1) and total FGFR1 (FGFR1) on serum-starved moEC left untreated or treated with recombinant L1-ΔTM (20 μg/ml) for 10 or 30 min. The blots were obtained from the same gel, the white line between the blots indicates the removal of intervening lanes. Right panel: FGFR1 phosphorylation in three biological replicates was quantitated by calculating the ratio between phospho-FGFR1 and total FGFR1. Data are normalized against the basal phosphorylation in untreated cells (indicated by the red dashed line). (**I**) moEC transduced with the empty vector (Vector) or with L1-Δ TM were subjected to tube formation assays in the presence of either the FGFR1 inhibitor PD173074 (PD) or DMSO as a control. For each analysis, data are expressed as means ± SEM from three independent experiments. Comparisons between experimental groups were done with two-sided Student's t-tests; **p<0.01, ***p<0.001.

DOI: https://doi.org/10.7554/eLife.44305.007

The following figure supplement is available for figure 3:

**Figure supplement 1.** Functional characterization of L1-ΔTM isoform and production of purified recombinant L1-ΔTM.
DOI: https://doi.org/10.7554/eLife.44305.008

*Javerzat et al., 2002*). To determine if L1-ΔTM function could be mediated by FGFR1, we first investigated whether soluble L1-ΔTM affects FGFR1 activation. As shown in *Figure 3H*, treating parental moEC with recombinant L1-ΔTM resulted in increased phospho-FGFR1, consistent with the L1-ΔTM-induced activation of FGFR1 signaling. Moreover, when L1-ΔTM-expressing moEC were subjected to tube formation assay in the presence of the small-molecule FGFR1 inhibitor PD173074 (*Skaper et al., 2000*), L1-ΔTM-dependent tube-forming activity was reduced to the level of control cells (*Figure 3I*). Thus, our data implicate FGFR1 signaling as an effector of L1-ΔTM in ECs.

## NOVA2 controls alternative splicing of L1-ΔTM in ECs

To gain further insights into the molecular mechanisms regulating the AS of *L1cam* in endothelium, we analyzed the sequence of mouse *L1cam* exon 25 and its flanking intronic regions, using SFmap (http://sfmap.technion.ac.il/) (*Paz et al., 2010*; *Akerman et al., 2009*) to search for putative binding sites of RNA-binding proteins. We sorted the results based on: i) the predicted ability of the RNA-binding protein to promote exon 25 skipping; ii) the presence of clusters of putative binding sites for a given RNA-binding protein, which are expected to enhance binding affinity; iii) the evolutionary conservation of the identified motifs; iv) the known expression of the identified factor in ECs. This analysis resulted in the identification of clustered and evolutionarily conserved putative binding sites for NOVA2, hnRNP A1 and SRSF3 (*Figure 4—figure supplement 1*), three factors previously reported to be expressed in ECs (*Giampietro et al., 2015*; *Holly et al., 2013*; *Lomnytska et al., 2004*).

To investigate the role of the identified candidate splicing factors in the AS of *L1cam*, we first performed a splicing assay in HeLa cells co-transfected with a minigene (p-L1) encompassing exons 24, 25, and 26 of *L1cam* along with the flanking intron sequences (*Figure 4A*) and the candidate splicing factors or with the empty vector. As shown in *Figure 4B* and *Figure 4—figure supplement 1*, skipping of *L1cam* exon 25 in the minigene was only observed upon overexpression of NOVA2, a key regulator of AS in ECs (*Giampietro et al., 2015*). In contrast, the overexpression of hnRNP A1 and SRSF3 had no effect on the skipping of exon 25, suggesting that the latter is a NOVA2-specific effect. To support a direct and specific role of NOVA2 in

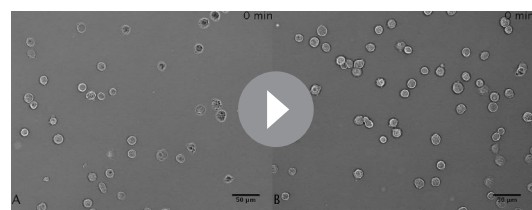

**Video 1.** L1-ΔTM promotes EC tube formation. Time-lapse videomicroscopy of tube formation on moEC transduced either with the empty vector (A) or with L1-ΔTM (B).
DOI: https://doi.org/10.7554/eLife.44305.009

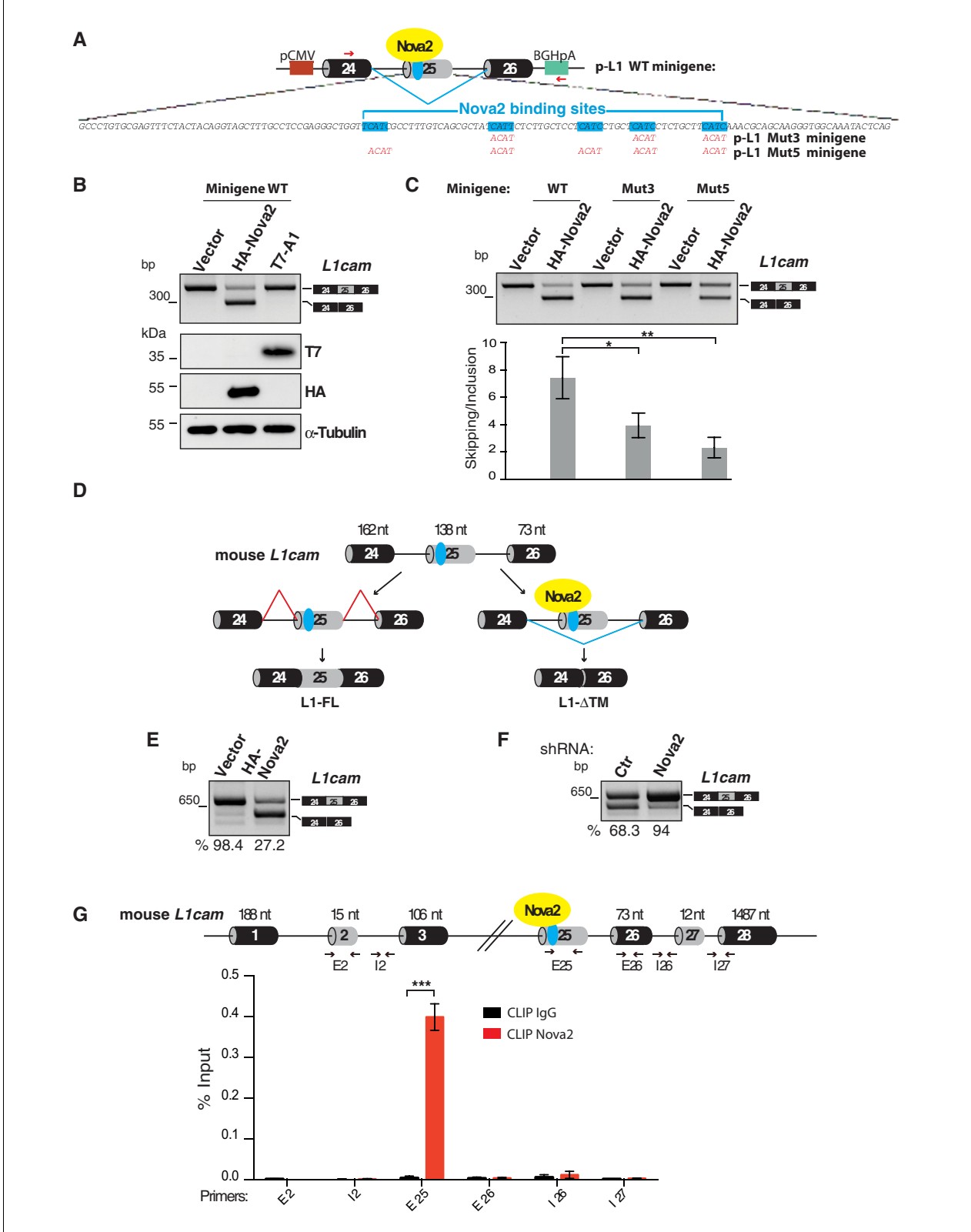

**Figure 4.** L1-ΔTM production is regulated by NOVA2 in ECs. (**A**) The mouse *L1cam* cassette, encompassing exons 24, 25 and 26, with the intervening intronic sequence was used to generate the p-L1 wild-type (WT) minigene. Mutated minigenes (Mut3 and Mut5) were generated by mutations in either three or all five NOVA-binding sites in exon 25 (YCAY repeats were mutated to ACAY). Boxes = exons; thin lines = introns; pCMV = promoter; BGHpA = polyadenylation sequence; red arrows indicate the annealing positions of the primers used for RT-PCR. (**B**) AS of transcripts of the WT
*Figure 4 continued on next page*

*Figure 4 continued*

minigene upon co-transfection of HeLa cells with either HA-NOVA2, T7-hnRNP A1 (T7–A1), or the empty vector. The ectopic expression of NOVA2 and hnRNP A1 was confirmed by western blotting with anti-HA and anti-T7 antibodies, respectively. (C) AS of transcripts from the WT and mutated minigenes in co-transfected HeLa cells. The histogram shows the ratio between skipping and inclusion of *L1cam* exon 25. Data indicate means ± SEM calculated from five independent experiments (n = 5). Tukey's multiple comparisons or two-sided Student's t-test were used for comparisons between experimental groups; *p<0.05; **p<0.01. (D) *L1cam* mouse genomic region comprising the AS exon 25 (grey box). Black boxes = constitutive exons; thin lines = introns; blue dot = YCAY cluster within exon 25 predicted to function as NOVA2-binding site. Bottom diagrams illustrate the inclusion (left) or the NOVA2-induced skipping of exon 25 (right). (E) AS of mouse *L1cam* exon 25 as determined by RT-PCR in moEC stably overexpressing HA-tagged *NOVA2* cDNA. (F) AS of mouse *L1cam* exon 25 in moEC transduced with an shRNA against *Nova2* or with a control shRNA (Ctr). The percentage of exon inclusion was calculated as described in Materials and methods and is shown below the gels. (G) CLIP was performed in moEC with anti-NOVA2 or control IgG. NOVA2-bound RNA was analyzed by RT-qPCR with *L1cam* primers E25 (annealing to the YCAY cluster), E2 (annealing to the exon 2), I2 (annealing to the intron 2), E26 (annealing to exon 26), I26 (annealing to the intron 26) and I27 (annealing to the intron 27 and exon 28). Binding of NOVA2 was calculated as % of input (see Materials and methods). Black arrows in the top diagram show the annealing position of the three primer sets. Data are expressed as means ± SEM calculated from three independent experiments (n = 3). ***p<0.001.
DOI: https://doi.org/10.7554/eLife.44305.010

The following figure supplements are available for figure 4:

**Figure supplement 1.** Evaluation of candidate SRFs on *L1cam* splicing.
DOI: https://doi.org/10.7554/eLife.44305.011

**Figure supplement 2.** *Nova2* expression levels and AS of *L1cam* in freshly purified mouse ECs.
DOI: https://doi.org/10.7554/eLife.44305.012

**Figure supplement 3.** Density-dependent expression and genetic manipulation of *Nova2* in mouse ECs and NOVA2-mediated splicing of *L1CAM* in human ECs.
DOI: https://doi.org/10.7554/eLife.44305.013

controlling *L1cam* AS, we mutated YCAY (Y = C/U) motifs, which represent putative binding sites for NOVA proteins (*Ule et al., 2006*), in *L1cam* exon 25 to ACAY, a sequence that reduces NOVA2 binding (*Jensen et al., 2000*). We found that mutations in only three repeats (Mut3) had a limited effect, whereas mutations in five repeats (Mut5) decreased skipping of *L1cam* exon 25 caused by NOVA2 overexpression (*Figure 4C*). These results are consistent with the dose-dependent binding of NOVA2 to its pre-mRNA targets (*Darnell, 2006*; *Leggere et al., 2016*) and further supported the involvement of NOVA2 in the AS regulation of *L1cam* exon 25. Such a hypothesis was also sustained by the following observations: i) the higher expression of NOVA2 in freshly purified ECs from mouse lung and in lu2EC as compared with total mouse lung or melanoma cell line B16, respectively, was accompanied by the skipping of *L1cam* exon 25 (*Figure 4—figure supplement 2* and *Figure 2—figure supplement 1*); and ii) *L1cam* exon 25 emerged as a novel NOVA2 target in ECs from the RNA-seq data in NOVA2-knockdown ECs (*Giampietro et al., 2015*) (*Supplementary file 2*) (see Materials and methods).

To investigate the causal relationship between NOVA2 expression and AS of the endogenous *L1cam*, we performed gain- and loss-of function studies in moEC (*Figure 4D–F* and *Figure 4—figure supplement 3*). In particular, forced expression of NOVA2 increased skipping of *L1cam* exon 25 (*Figure 4E* and *Figure 4—figure supplement 3*). Conversely, in NOVA2-depleted moEC (*Figure 4—figure supplement 3*) the skipping of *L1cam* exon 25 was markedly reduced (*Figure 4F*). The NOVA2-mediated AS regulation of *L1cam* exon 25 was also confirmed in lu2EC, another murine EC line (*Figure 4—figure supplement 3*).

Whether NOVA2 promotes exon skipping or inclusion depends on the location of its binding sites (i.e. YCAY clusters) in the pre-mRNA targets (*Ule et al., 2003*). In particular, NOVA2 usually induces exon skipping when bound to the exonic or upstream intronic region, while it stimulates exon inclusion when interacting with downstream intronic region. In the case of *L1cam* exon 25, the YCAY repeats are located within exon 25 (*Figure 4A*), consistent with the NOVA2-induced exon skipping observed in mouse ECs. Notably, the YCAY cluster is conserved between mouse and human *L1CAM* exons 25 with six repeats present in the human sequence (*Figure 4—figure supplement 3*). Accordingly, NOVA2 overexpression promotes skipping of *L1CAM* exon 25 also in human ECs (*Figure 4—figure supplement 3*).

To determine if NOVA2 directly regulates AS of the endogenous *L1cam*, we carried out UV cross-linking and immunoprecipitation (CLIP), which allows to identify direct protein-RNA interactions in live cells (*Ule et al., 2006*). RNA from UV cross-linked ECs was immunoprecipitated by using anti-

NOVA2 or control antibodies and then analyzed by RT-qPCR with primers spanning the YCAY cluster within *L1cam* exon 25. Primers that span either exon 26 or intron 26 were used as negative controls (*Figure 4G*). As shown in *Figure 4G*, NOVA2 bound to the endogenous *L1cam* transcript at the level of exon 25, while we observed no binding with either exon or intron 26. These data indicated a direct and specific interaction of NOVA2 with *L1cam* exon 25.

NOVA2 has been implicated also in the inclusion of exons 2 and 27 in neural cells (*Mikulak et al., 2012*). However, both RT-PCR and CLIP data showed no involvement of NOVA2 in the AS of these two exons in ECs (*Figure 4G* and *Figure 1—figure supplement 1*), further supporting the specific effect of NOVA2 on exon 25 in this cell type.

Collectively, our results support the notion that NOVA2 promotes skipping of *L1cam* exon 25 by binding to the YCAY motifs located within this exon.

## Clinical relevance of NOVA2-mediated AS of *L1CAM* in ovarian cancer vessels

We have recently described the expression of NOVA2 in vascular endothelium (*Giampietro et al., 2015*). Furthermore, our earlier reports demonstrated that L1CAM is expressed in tumor-associated vasculature (*Maddaluno et al., 2009*; *Magrini et al., 2014*). Taken together with the data presented here, these findings raise the hypothesis that NOVA2 regulates AS of *L1CAM* in cancer vessels. To test this possibility, we selected human ovarian carcinoma (OC) as a suitable model system. In fact, we found a markedly higher number of NOVA2-positive vessels in OC (identified via staining with the endothelial marker CD31) than in healthy ovaries (*Figure 5A* and *Figure 5—figure supplement 1*). The abundance and the vessel-restricted expression of NOVA2 in OC were also confirmed in tissue samples from the Human Protein Atlas project (https://www.proteinatlas.org/) (*Uhlén et al., 2015*) (*Figure 5—figure supplement 1*). The percentage of L1CAM-positive vessels was also dramatically increased in OC samples as compared to normal ovary (*Figure 5A*). In addition, NOVA2 was often co-expressed with L1CAM in OC vessels (*Figure 5B* and *Figure 5—figure supplement 2*). Thus, we applied RT-PCR to examine the AS of *L1CAM* in ECs isolated from OC (HOC-EC). As shown in *Figure 5C*, the L1-ΔTM isoform was readily detected in HOC-EC from seven independent OC samples. To test whether vascular L1-ΔTM in OC is associated with tumor angiogenesis, we measured the vessel density in a small cohort of OC samples pre-classified as *L1-ΔTM*-positive or negative by RT-qPCR (*Figure 5—figure supplement 2*). A significantly higher vessel density was found in *L1-ΔTM*-positive tumors (*Figure 5—figure supplement 2*). Furthermore, among the tumors which exhibited *L1-ΔTM* expression, the levels of *L1-ΔTM* correlated with vessel density ($r$ = 0.7671; $p<0.01$), measured by CD31 immunostaining (*Figure 5D*). These findings imply that the AS of *L1CAM* correlates with the degree of OC vascularization, which is consistent with a proangiogenic function of L1-ΔTM in this tumor type. To further assess the clinical relevance of our findings, we investigated the prognostic value of NOVA2 in OC, profiting from the RNA sequencing analysis of 372 OC patients performed through The Cancer Genome Atlas (TCGA) program. As shown in *Figure 5E*, higher expression of *NOVA2* correlated with shorter overall survival of the patients (HR: 1.486; $p=0.003$). Taken together, these results suggest that NOVA2 promotes AS of the *L1CAM* pre-mRNA in OC vessels, thus accounting for the vascular expression of L1-ΔTM, and highlight the proangiogenic role and the prognostic value of the NOVA2/L1-ΔTM axis in OC.

## Discussion

Our data implicated for the first time the splicing factor NOVA2 in the generation of a novel, EC-specific isoform of the cell adhesion molecule L1CAM, referred to as L1-ΔTM. Due to NOVA2-induced skipping of exon 25 that encodes the TM domain, L1-ΔTM is no longer associated to the cell surface and, hence, is released in the extracellular space. Consistent with the expression of NOVA2 in vascular ECs (*Giampietro et al., 2015*), the latter express and release high levels of L1-Δ TM.

We demonstrated that L1-ΔTM increases the ability of ECs to form tube-like structures in vitro and stimulates neovascularization in vivo. These data point to L1-ΔTM as a *bona fide* angiogenic factor which, however, belongs to a class of molecules highly divergent from the classic polypeptide growth factors that exert this function (vascular endothelial growth factors, fibroblast growth factors, etc.). To our knowledge, L1-ΔTM provides the first example of an immunoglobulin-like cell adhesion

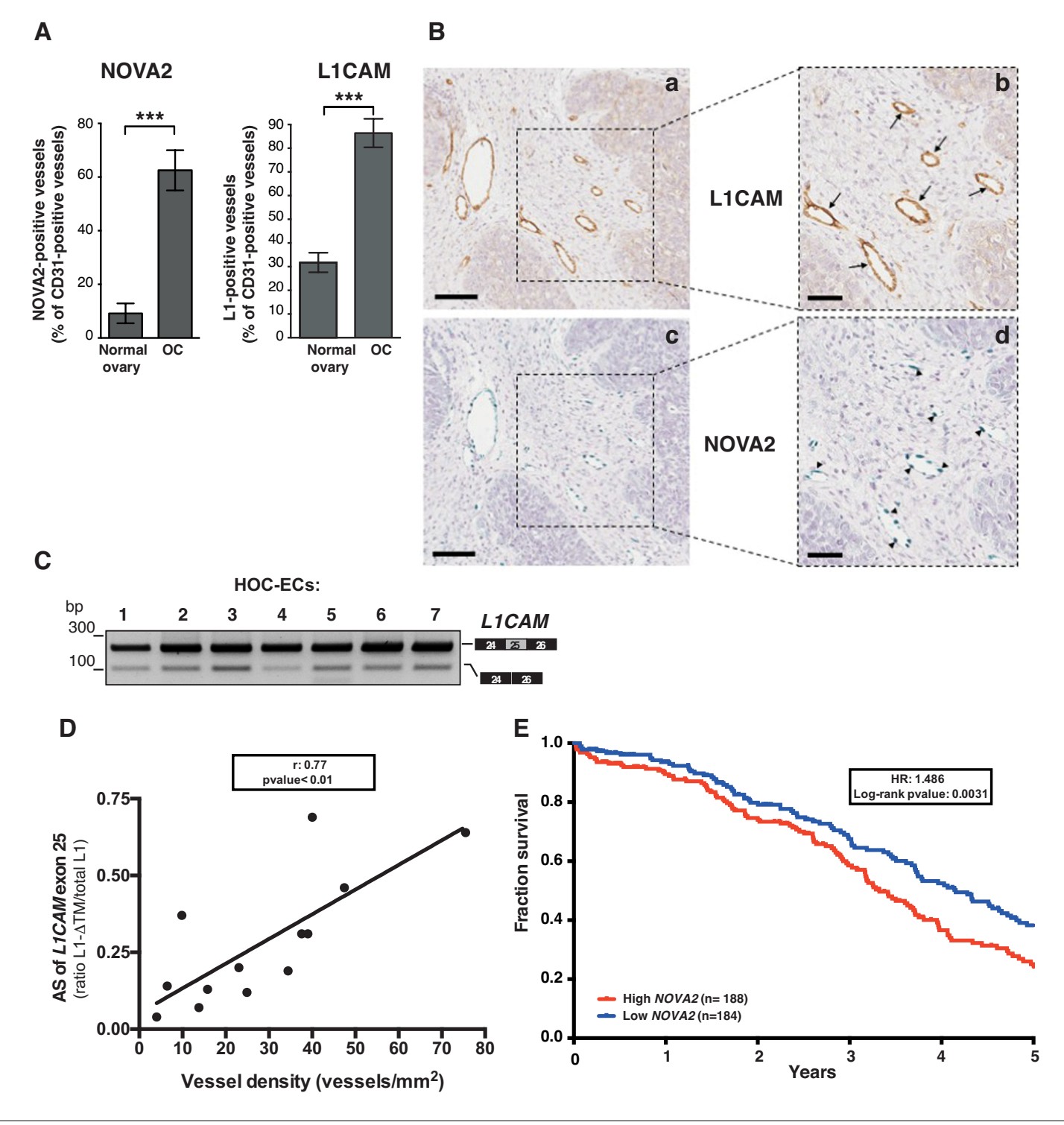

**Figure 5.** L1CAM and NOVA2 are co-expressed in ovarian cancer vessels. (**A**) Quantification of NOVA2- and L1CAM-positive vessels in healthy ovaries (n = 5) and high-grade serous ovarian cancer (OC) (n = 5). Data are expressed as the percentage of vessels positive for either NOVA2 (left) or L1CAM (right) over the total of CD31-positive vessels, and are shown as means ± SEM. Comparisons between experimental groups were done with two-sided Student's t-test; ***p<0.001. (**B**) Serial sections of OC samples were stained for L1CAM (brown) or NOVA2 (green). Arrows indicate L1CAM-positive vessels (panels **a** and **b**), and arrowheads indicate NOVA2 staining in EC nuclei (panels **c** and **d**). The right panels (**b, d**) show higher magnification of the areas delimited by dashed boxes. Scale bars: 100 µm, panels **a** and **c**; 50 µm, panels **b** and **d**. (**C**) AS of the human *L1CAM* exon 25 in seven EC batches purified and cultured from human ovarian carcinoma (HOC-ECs). (**D**) The skipping of *L1CAM* exon 25 was measured in OC samples (n = 13) as
*Figure 5 continued on next page*

*Figure 5 continued*

the ratio between L1-ΔTM and total *L1CAM* (determined by gel quantitation of RT-PCR products; see *Figure 5—figure supplement 2*). Microvessel density was measured in the same samples by CD31 staining and correlated with the AS of *L1CAM* exon 25. (E) Kaplan-Meier plot of overall survival in OC patients classified according to *NOVA2* expression (red curve, high expression; blue curve, low expression).

DOI: https://doi.org/10.7554/eLife.44305.014

The following figure supplements are available for figure 5:

**Figure supplement 1.** Expression of NOVA2 in OC and healthy ovarian vessels.

DOI: https://doi.org/10.7554/eLife.44305.015

**Figure supplement 2.** Co-localization of NOVA2 and L1CAM in OC vasculature and *L1CAM* splicing in OC samples.

DOI: https://doi.org/10.7554/eLife.44305.016

molecule converted into a soluble angiogenic factor through AS removal of the exonic sequence that encodes the TM domain. AS of the TM domain occurs in a broad spectrum of human genes (*Xing et al., 2003*), yet for most of them the biological significance and the functional outcome of such an event remain elusive. Based on our findings, it is conceivable that this post-transcriptional modification expands the repertoire of biologically active proteins that are released into the extracellular compartment.

We discovered that ECs acquire angiogenic properties not only upon ectopic expression of L1-Δ TM, but also when exposed to exogenously added L1-ΔTM (either as CM or as a recombinant protein). This supports the notion that extracellular L1-ΔTM can promote angiogenesis via both autocrine and paracrine stimulation. The latter mode of action, in particular, might represent a way to amplify the angiogenic signal provided by vascular L1CAM. Indeed, in most cases only a fraction of vessels within tumor and inflammatory tissues (i.e. the most prominent conditions where vascular L1CAM is detected), exhibits L1CAM expression (*Issa et al., 2009*; *Kaifi et al., 2006*; *Maddaluno et al., 2009*; *Magrini et al., 2014*). Thus, the AS-mediated generation of L1-ΔTM, and hence its release, would contribute to spread the proangiogenic stimulus also to L1CAM-negative vessels.

In different cellular contexts, such as neurons, L1CAM frequently engages in homophilic L1CAM-L1CAM interactions between adjacent cells, which underlie its function in cell-cell adhesion (*Maness and Schachner, 2007*). However, our experiments were conducted in two EC lines, moEC and luEC, that express no or very little endogenous L1CAM. This implies that, regardless of autocrine or paracrine activity, L1-ΔTM stimulates ECs by heterophilic interactions with different surface molecules. Accordingly, the extracellular portion of L1CAM has been reported to interact with a wide spectrum of membrane proteins, including different integrins, CD24, NCAM, F11R, neuropilins, etc. (*Haspel and Grumet, 2003*). With regard to potential L1-ΔTM interactors, one of the most intriguing aspects of our study is related to the crosstalk between L1-ΔTM and FGFR1. Following the pioneering studies of Doherty and Walsh in neuronal systems (*Williams et al., 1994*), L1CAM has been proposed to interact with the FGFR signaling machinery in different experimental settings (*Mohanan et al., 2013*; *Son et al., 2011*; *Zecchini et al., 2008*), possibly entailing a direct binding between the two molecules (*Kulahin et al., 2008*). Our data show for the first time a functional interplay between L1-ΔTM and FGFR1 in ECs, and implicate it in L1-ΔTM-induced angiogenesis. Future studies should elucidate whether interfering with the L1-ΔTM/FGFR1 crosstalk will open novel perspectives for antiangiogenic therapy.

In neural cell types, NOVA2 is required for the inclusion of the *L1CAM* exons 2 and 27 (*Mikulak et al., 2012*). This is not the case in ECs, where even upon ectopic over-expression of NOVA2 exons 2 and 27 are not included in the *L1CAM* mRNA (*Figure 1—figure supplement 1*). Moreover, despite the high expression of NOVA2 in the nervous system (*Yano et al., 2010*; *Giampietro et al., 2015*), we did not detect L1-ΔTM in human or mouse brain (*Figure 1*), which implies that NOVA2 is not involved in the AS of exon 25 in neural cells. Interestingly, our CLIP data on ECs revealed the binding of NOVA2 to exon 25, while no binding was detected with the regions flanking the neural-specific exons 2 and 27 (*Figure 4G*). Overall, these observations point to differential and cell type-specific interactions of NOVA2 with the *L1CAM* pre-mRNA that, in turn, underlie distinct AS outcomes. Future studies should aim at identifying the molecular determinants of such a cell type specificity.

NOVA2 is significantly up-regulated in OC vasculature as compared to vessels in healthy ovary. Of note, NOVA2 expression is detected only in the vascular endothelium of OC and, in particular, in the nucleus of ECs. It is remarkable that, in spite of such a restricted expression pattern, NOVA2 has a prognostic value in OC, given its association with shorter patients' survival. To the best of our knowledge, NOVA2 is the only SRF reported to be upregulated in cancer vasculature, including OC (this study) and colorectal carcinoma (*Gallo et al., 2018*). This implies that NOVA2-mediated AS might play a relevant, yet underappreciated, role in the phenotypic and functional aberrancies of the tumor vessels (*Carmeliet and Jain, 2011*). Consistent with the proposed role of the NOVA2/L1-ΔTM axis in tumor angiogenesis, increased NOVA2 levels are frequently accompanied by the expression of L1-ΔTM in tumor vessels. Furthermore, L1-ΔTM expression correlates with OC vascularization, although further studies should investigate the causal role of L1-ΔTM in tumor angiogenesis. These data, together with our observations on the role of vascular L1CAM in tumor angiogenesis and progression (*Magrini et al., 2014*) and on the proangiogenic activity of L1-ΔTM, point to NOVA2 as a potential driver of OC neovascularization. In addition, our results further strengthen the rationale for testing vascular L1CAM as a novel target for antiangiogenic strategies. Indeed, interfering with the function of cell surface-associated endothelial L1CAM (*Magrini et al., 2014*) and, at the same time, with that of extracellular L1-ΔTM might result in efficient repression of cancer angiogenesis.

The translational implications of our findings extend beyond the therapeutic area. Based on its specific up-regulation in OC vessels (and likely in other cancer types as well), we speculate that L1-ΔTM could serve as a new circulating biomarker of cancer-associated vasculature. In this case, it may become necessary to discriminate between vessel-derived L1-ΔTM and the ectodomain of transmembrane L1CAM which is released by various cell types, including tumor cells, upon proteolytic cleavage (*Kiefel et al., 2012*; *Yu et al., 2016*). Assays aimed at detecting the cytosolic domain of circulating L1CAM would serve this purpose. Circulating L1-ΔTM could be harnessed for diagnostic and prognostic purposes, ranging from the detection of primary and recurrent OC to monitoring the therapeutic response, all objectives that would improve dramatically the clinical management of OC patients. Along this line, the circulating levels of vessel-derived L1-ΔTM could offer a novel predictive tool to identify patients eligible for antiangiogenic therapies.

In summary, we provide evidence that L1-ΔTM is a novel, soluble isoform of L1CAM generated in ECs through NOVA2-mediated AS. L1-ΔTM exhibits proangiogenic activity and is up-regulated in tumor vessels, which may have high translational and clinical relevance.

## Materials and methods

### Cell culture

Human cervix carcinoma HeLa cells (ATCC, CCL-2) were grown in Dulbecco's modified Eagle's medium (DMEM, Euroclone) supplemented with 10% fetal bovine serum (FBS, Euroclone) and 2 mM L-glutamine (Lonza). Human embryonic kidney (HEK) 293 cells (FreeStyle 293 F Cells, Gibco) were grown in FreeStyle 293 Expression Medium (Gibco). N-acetylglucosaminyltransferase I-deficient HEK293 cells [GnTI(-)] (*Reeves et al., 2002*) were grown in FreeStyle 293 Expression Medium supplemented with 0.1% Pluronic F-127 (Sigma-Aldrich) and 1% FBS.

Vascular endothelial (VE) cadherin-positive ECs, here referred to as moEC, have been described in *Lampugnani et al. (2002)* and *Taddei et al. (2008)*. An extensive characterization confirmed the endothelial nature of these cells (*Lampugnani et al., 2003*; *Taddei et al., 2008*). Mouse lung-derived luEC and lu2EC were described in *Magrini et al. (2014)*, and *Bazzoni et al. (2005)*, respectively. Primary EC were isolated from mouse lung with anti-CD31 immunomagnetic beads as described (*Malinverno et al., 2017*). More than 99% isolated cells were positive for VE-cadherin and CD31 by immunofluorescence (*Figure 4—figure supplement 2*). Furthermore, isolated cells exhibited a marked enrichment in the mRNA for VE-cadherin and CD31, while no or very little expression was detected for epithelial (E)-cadherin and alpha-smooth muscle actin (*Figure 4—figure supplement 2*).

Mouse EC lines derived from whole embryo (EmbEC), lu2EC and moEC were cultured in DMEM-High Glucose (Lonza) with 10% FBS, 2 mM L-glutamine, 100 U/L penicillin/streptomycin (Sigma-Aldrich), 1 mM sodium pyruvate (Sigma-Aldrich), 25 mM HEPES (Sigma-Aldrich), 100 μg/ml heparin (from porcine intestinal mucosa; Sigma-Aldrich) and 50 μg/ml EC growth supplement (ECGS from

bovine pituitary gland; Sigma-Aldrich). LuEC were cultured in MCDB131 medium (Sigma-Aldrich) supplemented with 20% FBS, 2 mM L-glutamine, 1 mM sodium pyruvate (Gibco), 100 μg/ml heparin and 50 μg/ml ECGS as previously described (*Magrini et al., 2014*). To enhance EC adhesion, plates were coated with 0.1% porcine gelatin (Difco) and incubated overnight at 37°C before seeding. Human umbilical vein EC (HUVEC, isolated as in (*Giampietro et al., 2015*) and HUVEC/TERT2 (Ever-cyte) were cultured on porcine gelatin-coated plates in MCDB131 containing 20% FBS and the same supplements as moEC and luEC.

Human ovarian cancer-derived EC (HOC-EC) have been described previously. HOC-EC cultures were found to contain at least 99% endothelial marker-positive (LDL uptake, CD31, von Willebrand Factor) cells, and <1% alpha-smooth muscle actin-positive cells (*Ghilardi et al., 2015*).

Human immortalized cerebral microvascular ECs (hCMEC/D3) were provided by PO Couraud and cultured as described previously (*Weksler et al., 2013*). hCMEC/D3 cells were grown on Collagen type I coated dishes (Corning).

Mouse melanoma B16F10 cells (hereafter referred to as B16) were purchased from ATCC and cultured in DMEM, 10% FBS.

All cells were routinely tested for mycoplasma with a PCR-based method. Whenever applicable, cell line authentication was performed with the GenePrint STR System typing kit (Promega).

## Plasmids

The cDNAs encoding HA-tagged, full-length mouse *L1cam* (deleted of the neural-specific exons 2 and 27) and its TM—deleted *L1cam* isoform (deleted of exons 2, 25 and 27) (*Figure 1—figure supplement 1*) were generated by PCR-mediated mutagenesis of pcDNA3.1-Hygro/L1 construct (*Magrini et al., 2014*) and then cloned into the pcDNA3.1-Hygro(-) (Invitrogen) or in the pLenti-III-HA (ABM Inc) by using standard DNA cloning procedures. To generate the *p-L1* WT minigene (*Figure 4*), the genomic mouse *L1cam* cassette was amplified with primers p-L1_F/R and then cloned into the EcoRI and XbaI restriction sites of pcDNA3.1(+) (Invitrogen). In p-L1 Mut minigenes (Mut3 and Mut5), the NOVA2-binding sites TCAT were replaced with ACAT by PCR-mediated mutagenesis of p-L1 WT minigene. HA-tagged human *NOVA2* cDNA was PCR amplified with primers pcDNA-NOVA2_F/R and cloned into the BamH1 and XhoI restriction sites of the pcDNA3.1(+), whereas T7-tagged hnRNP A1 expression vector was generated as described previously (*Bonomi et al., 2013b*). The expression vector for T7-SRSF3 was kindly provided by J. Caceres (MRC Human Genetics Unit, University of Edinburgh, UK).

To produce recombinant L1-ΔTM protein (see below), the *L1-ΔTM* cDNA was amplified with pUPE-L1_F/R primers from pcDNA Hygro-L1-ΔTM. The resulting amplicon, bearing in-frame 5'-BamHI and 3'-NotI restriction sites was then sub-cloned into pUPE.06.45 mammalian expression vector (U-Protein Express B.V., Utrecht, The Netherlands), bearing a cystatin signal peptide followed by N-terminal 6xHis, 3xStrepII tags, and a specific cleavage site for Tobacco Etch Virus protease (TEV) preceding the 5'-BamHI restriction site. All PCR products were verified by sequencing, whereas all primers are listed in *Supplementary file 1*.

## Cell transfection

HeLa cells were transiently transfected with Lipofectamine 3000 (Invitrogen), according to the manufacturer's protocol. luEC were stably transfected with pcDNA-Hygro/L1-FL or pcDNA-Hygro/L1-ΔTM or the empty vector pcDNA-Hygro(-) (see below), followed by the selection of positive clones as previously described (*Magrini et al., 2014*). HEK293 and HEK293 GntI(-) cells were transfected using polyethyleneimine (Polysciences, Germany) as described (*Durocher et al., 2000*).

## Lentivirus production and transduction

To obtain viruses expressing L1CAM isoforms, mouse *L1cam* cDNAs were cloned in the pLenti-III-HA vector (ABM Inc, see above). Lentiviruses were produced using HEK293T as recipient cells (ATCC, CRL-1573). Cells were co-transfected with 10 μg of each packaging vector pMD2.G (Addgene, plasmid #12259), pRSV-Rev (Addgene, plasmid #12253), pMDLg/pRRE (Addgene, plasmid #12251) and 10 μg of L1CAM-expressing lentiviral vectors by calcium phosphate precipitation method. After 24 hr, the medium containing the lentiviruses was filtered, supplemented with 8 μg/ml of polybrene (Sigma-Aldrich) and used to infect moEC.

MoEC were transduced with HA-NOVA2 or with shRNA vectors as described in (*Giampietro et al., 2015*), using lentiviral vectors carrying human HA-tagged *NOVA2* cDNA (pLenti-GIII-CMVhumanNOVA2-HA, THP Medical Products) or shRNA for the mouse *Nova2* gene (GIPZ shRNAs from Open Biosystems), respectively. After 48 hr of infection, the medium was refreshed and puromycin selection (3 µg/ml) was started. Since NOVA2 expression is regulated by EC density (*Giampietro et al., 2015*), for the analysis of *L1CAM* splicing NOVA2-knockdown moEC were used as confluent monolayers (500000 cells in 35 mm Petri dishes), whereas moEC overexpressing HA-tagged NOVA2 were tested at low density (500000 cells in 100 mm Petri dishes). hCMEC/D3 cells were transduced with lentiviral vectors carrying human HA-tagged *NOVA2* cDNA (pLenti-GIII-CMVhumanNOVA2-HA, THP Medical Products) and after 48 hr, infected cells were selected with 3 µg/ml puromycin for 5 days.

## RNAi

To ablate *Nova2* in lu2EC, we used siRNA from Sigma-Aldrich (MISSION siRNA ID SASI_Mm01_00094763) and the corresponding negative control (MISSION siRNA Universal Negative Control #1). Transfection was performed with Lipofectamine RNAiMax (Invitrogen) in accordance with the manufacturer's instructions. Two subsequent transfections (with 24 hr intervals) were performed with 13 nM siRNA, and cells were collected 24 hr after the second transfection. To knock down *NOVA2* in HUVEC/TERT, we used siRNA from Sigma-Aldrich (MISSION siRNA ID SASI_Hs01_00220812) and the corresponding negative control (MISSION siRNA Universal Negative Control #1). Transfection was performed with Lipofectamine RNAiMax (Invitrogen) following the manufacturer's instructions. Two subsequent transfections (with 24 hr intervals) were performed with 30 nM siRNA, and cells were collected 48 hr after the second transfection.

## Morpholino treatment

Subconfluent (80–85%) lu2EC cells were transfected with the MO-L1-SB oligonucleotide (5'- CCTG TACATTTTCTAGGTTACCTGA-3'; GENE-TOOLS) at 15 µM plus 8 µM Endo-Porter PEG system (GENE-TOOLS) according to the manufacturer's instructions. An irrelevant morpholino oligonucleotide (Standard Control Oligo, GENE-TOOLS) was used as control. After 16 hr, lu2EC were washed three times with PBS and starved for 24 hr in the following medium: DMEM High Glucose, 2 mM L-glutamine, 100 U/l penicillin streptomycin, 1 mM sodium pyruvate, 25 mM HEPES, 0.5% FBS. Conditioned media (CM) were collected and centrifuged for 10 min to remove cell debris.

## Immunoblot analysis

Total protein extraction and immunoblot were performed as previously described in (*Magrini et al., 2014*). Briefly, total proteins were extracted by solubilizing cells in Laemmli buffer (4% SDS, 16% glycerol, 40 mM Tris-HCl pH 6.8). To ensure an equal loading, lysates were quantified using Pierce BCA protein assay kit (Thermo Fisher Scientific) and 20 µg loaded on gel. The amount of EC-derived CM analyzed by immunoblotting was normalized against the number of producing cells. Lysates and CM were separated using SDS–PAGE and analyzed by western blotting. The following primary antibodies were used: anti-NOVA2 C-16 (1:200; Santa Cruz Biotechnology), anti-α-Tubulin (1:50,000; Sigma-Aldrich), anti-HA High Affinity (1:1000; Roche), anti-T7 tag (1:5000; Novagen), anti-Vinculin (1:5000 Millipore), anti-L1CAM [1 µg/ml; (*Magrini et al., 2014*)], anti-L1CAM cytoplasmic domain [1:8000; Ral1cd, kindly provided by V Lemmon; (*Schaefer et al., 2002*)]; anti-FGFR1 (1:200; clone M2F12, Santa Cruz Biotechnology) and anti-phospho-FGFR1(Y653/654) (1:1000 Santa Cruz Biotechnology). The following secondary antibodies conjugated to horseradish peroxidase (Jackson ImmunoResearch) were used: anti-Mouse (1:5000), anti-Goat (1:5000), anti-Rat (1:5000) and anti-Rabbit (1:10000). Immunostained bands were detected using the chemiluminescent method (Clarity ECL Western Blotting Substrate, Bio-Rad) and images were obtained by ChemiDoc Imaging Systems (Bio-Rad).

## Immunofluorescence

ECs were grown on gelatin-coated coverslips and then fixed with 4% PFA for 10 min at room temperature (RT). Membrane permeabilization was obtained incubating coverslips in ice-cold PBS and 0.5% Triton X-100 for 3 min at 4°C. Cells were then incubated for 1 hr at RT in a humid chamber

with blocking solution (PBS, 2% BSA, 5% donkey serum, and 0.05% Triton X-100). Samples were then incubated for 2 hr with primary antibodies (anti-L1CAM, 1 µg/ml) diluted in blocking buffer, followed by the incubation with Cy3 donkey anti-Rabbit (1:600, Listarfish) secondary antibodies (45 min at RT). Finally, samples were washed in PBS and nuclei counterstained with DAPI solution (0.2 µg/ml, Sigma-Aldrich). Confocal microscopy was performed with a Leica SP2 confocal microscope equipped with a motorized stage and violet (405 nm laser diode), blue (488 nm Argon), yellow (561 nm laser diode), and red (633 nm HeNe laser) excitation laser lines. For quantification purposes, at least five fields for each condition were counted and the average of the positive/negative cells was calculated. The statistical difference among groups was determined as described in the Statistical analysis section.

## Immunohistochemistry

Fresh tissue samples were obtained upon informed consent from patients undergoing surgery at the Gynecology Division of the European Institute of Oncology (Milan). Sample collection was performed under the protocol n. R789-IEO approved by the Ethics Committee of the European Institute of Oncology. The immunohistochemical analysis of L1CAM expression was carried out on a panel of high-grade serous ovarian carcinoma. Fresh samples were 4% PFA-fixed and paraffin embedded. After an overnight at 37°C, tumor sections (3 µm) were deparaffinized using Leica ST5020 Multistainer. Tissue sections were treated with the antigen unmasking solution EDTA pH 8 in pre-warmed water bath at 95°C for 50 min and then endogenous peroxidases were blocked using 3% Hydrogen peroxide solution (Carlo Erba). Tissue sections were incubated in blocking solution (TBS, 4% BSA, 0.05% Triton X-100) for 1 hr. Samples were then incubated for 2 hr with primary antibodies (anti-L1CAM, 1 µg/ml; anti-NOVA2 C-16, 1:100 SantaCruz; anti-CD31, 1:50 Abcam) followed by secondary antibodies incubation (Dako EnVision + System HRP Labelled Polymer or Goat-on-Rodent HRP Polymer from Biocare) for 30 min at RT. Dako chromogen substrate (Liquid DAB +Substrate Chromogen System Ref. K3468) or Vina Green Chromogen Kit (Biocare Medical) were used for signal detection. Samples were counterstained using Hematoxylin solution (Leica). Double immunohistochemistry for NOVA2 and L1CAM (using the same primary antibodies as above) was performed as previously described (*Giampietro et al., 2015*). Pictures of stained sections were acquired with the Aperio ScanScope XT instrument. For quantification of CD31 and NOVA2 staining, five different fields were counted for each section from five independent samples of normal ovaries or ovarian cancer. Vessel density was determined calculating the number of CD31-positive vessels per area unit ($mm^2$) using a proprietary tool of the Aperio ImageScope software (Leica Biosystems Imaging).

## RNA extraction, RT-PCR, RT-qPCR

Total RNA was isolated from both cultured cells and paraffin-embedded samples. RNA from cultured cells was obtained by using the RNeasy Mini Kit (QIAGEN), while tissue RNA was extracted with AllPrep DNA/RNA FFPE (QIAGEN) according to manufacturer's instructions. cDNA was obtained starting from 500 to 1000 ng of total RNA with Superscript IV RT cDNA synthesis kit (Invitrogen) according to the manufacturer's instructions and an aliquot (1–2 µl) of cDNA was then PCR-amplified (with GoTaq DNA Polymerase, Promega). The percentage of exon inclusion was calculated as the ratio between the intensity of the band of *L1cam* transcripts with the exon included and the total intensity of the *L1cam* bands. Splicing of *L1CAM* in human normal tissues (brain, heart, liver, kidney, lung and trachea, Clontech), normal human ovary and breast samples (Ambion) and human EC lines (HUVEC and hCMEC/D3) was analyzed with primers hL1E23_F and hL1E28_R (*Supplementary file 1*). RT-PCR analysis on FFPE samples of ovarian cancer tissue was performed with the primers hL1E24_F and hL1E26_R (*Supplementary file 1*) after selection of suitable areas of the tumors by a trained pathologist (GB). Splicing of *L1cam* in mouse tissues (brain, cortex, testis, lung, tongue and liver), freshly purified ECs from mouse lung and EC lines was analyzed with primers mL1E23_F and mL1E28_R (*Supplementary file 1*). Mouse tissues were obtained from Karolinska Institutet (Stockholm, Sweden) and IRCCS San Raffaele Scientific Institute (Milan, Italy), in accordance to Institutional Animal Care and Use Committees. Band intensity on agarose gel was quantified with the NIH Image J program (version 1.50i). All PCR products were verified by sequencing. For RT-qPCR experiments, cDNA samples were amplified with QuantiTect SYBR Green PCR (QIAGEN) by

using LightCycler 480 (Roche). Target transcript levels were normalized to those of *GAPDH* or *Ubb* housekeeping genes. All primers used in RT-qPCR are listed in *Supplementary file 1*.

## Crosslinking and immunoprecipitation (CLIP)

CLIP assay was performed as previously described (*Paronetto et al., 2014*; *Paronetto et al., 2011*). moEC were irradiated once with 150 mJ/cm$^2$ in a Stratlinker 2400 at 254 nm. Cell suspension was centrifuged at 4000 rpm for 3 min, and pellet was incubated for 10 min on ice in lysis buffer [50 mM Tris-HCl, pH 7.4, 100 mM NaCl, 1% Igepal CA-630 (Sigma-Aldrich), 0.1% SDS, 0.5% sodium deoxycholate, 0.5 mM Na$_3$VO$_4$, 1 mM DTT, protease inhibitor cocktail (Sigma-Aldrich), and RNase inhibitor (Promega)]. Samples were briefly sonicated and incubated with 10 µl of 1:1000 RNase I (100 U/µL, Ambion) dilution and 2 µl of DNase (2 U/µL, Ambion) for 3 min at 37°C shaking at 1100 rpm, and then centrifuged at 15,000 g for 10 min at 4°C. One milligram of extract was immunoprecipitated using anti-NOVA2 antibody (C-16, Santa Cruz Biotechnology) or purified IgG (negative control) in the presence of protein A/G magnetic Dynabeads (Life Technologies). Immunoprecipitates were incubated overnight at 4°C under constant rotation. After stringent washes with high salt buffer (50 mM Tris-HCl, pH 7.4, 1 M NaCl, 1 mM EDTA, 1% Igepal CA-630, 0.1% SDS, 0.5% sodium deoxycholate), beads were equilibrated with PK buffer (100 mM Tris-HCl, pH 7.4, 50 mM NaCl, 10 mM EDTA). An aliquot (10%) was kept as input lysate, while the rest was treated with 50 µg Proteinase K and incubated for 20 min at 37°C shaking at 1100 rpm. 7 M urea was added to the PK buffer and incubated for further 20 min at 37°C shaking at 1100 rpm. The solution was collected and phenol/CHCl$_3$ (Ambion) was added. After incubation for 5 min at 30°C shaking at 1100 rpm, phases were separated by centrifuging for 5 min at 13000 rpm at RT. The aqueous layer was transferred into a new tube and precipitated by addition of 0.5 µl glycoblue (Ambion), 3 M sodium acetate pH 5.5 and 100% ethanol. After mixing, the solution containing retained RNA was precipitated overnight at −20°C. RNA extracted from both the input material and the immunoprecipitates was then analyzed by RT-qPCR as described in the above paragraph. The binding was expressed as percentage of the input material.

## Minigene splicing assay

HeLa cells were transiently co-transfected by using Lipofectamine 3000 with 500 ng of *p-L1 WT* minigene and either 500 ng of protein expression vectors (HA-NOVA2, T7-hnRNPA1 and T7-SRSF3) or the empty vector. Five-hundred ng of each mutated minigene (Mut3 and Mut5) were transfected with 250 ng of HA-NOVA2 expression vector plus 250 ng of the empty vector; as a control, we used 500 ng of the empty vector. Total RNAs were extracted from HeLa cells after 24 hr and analyzed by RT-PCR with primers p-L1_F and BGH_R annealing to mouse *L1cam* exon 25 and to the Bovine Growth Hormone (BGH) polyadenylation site, respectively. Primers are listed in *Supplementary file 1*.

## L1-ΔTM recombinant protein production

For the production of secreted, recombinant L1-ΔTM (*Figure 2—figure supplement 1*), FreeStyle 293 F cells were cultured and transfected in FreeStyle medium. Four hours after transfection with pUPE.06.45-L1-ΔTM, Primatone RL (Sigma-Aldrich) was added to the culture medium at the final concentration of 0.6%. The culture medium was harvested 7 days after transfection by 10 min centrifugation at 1000 g. The supernatant containing secreted L1-ΔTM was loaded at 0.5 ml/min on a 1 ml StrepTrap HP column (GE Healthcare). The column was washed with 10 column volumes (cv) of buffer P, composed of 50 mM HEPES and 250 mM NaCl pH 7.5. Elution was performed with 1 cv of the same buffer supplemented with 5 mM d-Desthiobiotin (Sigma-Aldrich). After elution, protein was incubated overnight at 4°C with TEV protease to remove the N-terminal affinity tags. The resulting protein was concentrated by centrifugation using Vivaspin Turbo 15, 100000 MWCO centrifugal filters (Sartorius) to a volume less than 500 µl and then loaded onto a Superdex 200 10/300 GL (GE Healthcare) gel filtration column equilibrated with buffer P. The elution peak corresponding to L1-ΔTM as judged by SDS-PAGE analysis was collected and concentrated to 1 mg/ml. The purified sample was then flash-frozen in liquid nitrogen and stored at −80°C until further usage.

## Conditioned medium production

Confluent mouse ECs were washed three times with PBS 1x and cultured for 48 hr in the following medium: DMEM High Glucose, 2 mM L-glutamine, 100 U/l penicillin streptomycin, 1 mM sodium pyruvate, 25 mM HEPES, 0.5% FBS. Conditioned media (CM) were collected and centrifuged for 10 min to remove cell debris.

## In vitro tube formation assay

A Matrigel-based tubulogenesis assay was performed to assess the ability of ECs to form an organized capillary-like network. Before proceeding with the assay, 96-well plate was coated with 50 μl/well of Growth Factor-Reduced Matrigel (BD Biosciences) and left for 1 hr at 37°C for gelification. To assess the cell-autonomous effect of L1-ΔTM, transduced moEC were plated on Matrigel-coated plates in complete growth medium. Tube-like structures were manually counted under the microscope after 24 hr.

To assess the paracrine effect of L1-ΔTM, confluent moEC were cultured overnight in the same medium used for CM production. The day after, moEC were seeded on polymerized Matrigel-coated wells in the appropriate CM or, where indicated, were treated with different concentrations of recombinant, purified L1-ΔTM. After 8 hr of incubation at 37°C, tubes-like structures were counted under the microscope.

To block L1-ΔTM activity, transduced moEC were pre-incubated for 1 hr at 37 °C with 10 μg/ml of an anti-L1CAM blocking antibody [clone 324 (*Di Sciullo et al., 1998*; *Appel et al., 1993*)] or with control rat IgG (Sigma-Aldrich). Cells were then seeded on Matrigel-coated wells in complete growth medium containing 10 μg/ml of anti-L1CAM 324 (or control rat IgG) and tube-like structures were counted manually 8 hr later under the microscope. Where indicated, transduced moEC were pre-incubated for 1 hr at 37 °C with FGFR inhibitor PD173074 at a final concentration of 70 nM. Cells were then seeded on Matrigel-coated wells in normal growth medium containing 70 nM of PD173074 and tube-like structures were counted manually after 8 hr under the microscope. Both image acquisition and cell counts were performed using EVOS FL Imaging System. Cells were plated in triplicate (technical replicates) and the experiment was performed three times (biological replicates).

## Matrigel plug

All animal studies were performed following a protocol approved by the fully authorized animal facility of European Institute of Oncology and by the Italian Ministry of Health (as required by the Italian Law) (IACUC n. 1256/2015) and in accordance to EU directive 2010/63. The sample size estimation was based on previous studies and pilot experiments. C57Bl/6 mice were injected into the right flank with 200 μl of CM derived from mouse ECs expressing either L1-ΔTM or L1-FL or from control ECs in a final volume of 600 μl of Growth Factor-Reduced Matrigel (BD Biosciences). Matrigel containing 0.5 μg of FGF2 was used as positive control. Groups were composed by three mice for each construct. Plugs were removed 7 days after injection, fixed in 4% PFA and paraffin embedded. Sections from fixed plugs were stained for CD31 as described above. The number of CD31-positive vessels that invaded Matrigel plugs were evaluated by manual counting of five different fields per section using Axioskop two microscope (Leica Biosystems).

## Survival analysis

To determine the prognostic relevance of NOVA2, overall survival curves of ovarian cancer patients were built analyzing the GDC TCGA dataset with the UCSC Xena web tool (http://xena.ucsc.edu/). Survival plots were drawn using the Kaplan-Meier method and patients were stratified according to NOVA2 expression using median as threshold value. The log-rank Mantel-Cox test was employed to determine any statistical difference between the survival curves of the cohorts.

## Analysis of RNA-seq data from NOVA2-knockdown ECs

We previously performed RNA sequencing on NOVA2-knockdown moEC versus their parental control (*Giampietro et al., 2015*). To capture additional NOVA2-regulated AS events in that experimental system, we first aligned each of the four samples independently (two control and two knockdown replicates) with the Vertebrate Alternative Splicing and Transcription Tools (*vast-tools*) (*Tapial et al.,*

*2017*) (https://github.com/vastgroup/vast-tools). Then both replicates of each experimental point were pooled with *vast-tools merge* using the default parameters. This increased markedly the read coverage at each of the exon-exon junctions per experimental condition. Then, we applied *vast-tools compare* to perform a differential splicing analysis between the pooled control and the pooled knockdown samples, using a cutoff of delta percent spliced in ($\Delta$PSI) of 15 with default settings.

### Statistical analysis and data reporting

Independent experiments were considered as biological replicates. When performed, technical replicates deriving from the same biological replicate were averaged. For in vivo experiments, each mouse represented one biological replicate. For staining of human tissues, each patient represented one biological replicate. Data are expressed as mean ±SEM, calculated from at least three independent experiments. Student's two-tailed t test or ANOVA multiple comparison test, followed by Tukey's post hoc analysis, were used to compare two or three or more groups, respectively, and to determine statistical significance (GraphPad Prism 5). The correlation between the expression of L1-$\Delta$TM and tumor vessel density was assessed using the Spearman rank correlation coefficient.

Differences were considered significant at $p < 0.05$. Asterisks correspond to p-value calculated by two-tailed, unpaired, t-test (*$p < 0.05$, **$p < 0.01$, ***$p < 0.001$). The sample size estimation was based on previous studies and pilot experiments.

## Acknowledgements

This paper is dedicated to the memory of our friend and colleague Prof. Giovanni Morrone. We are grateful to F Bianchi for gene expression analysis, S Pirroni for assistance with RNA extraction, A Canciani for assistance with L1-$\Delta$TM cloning, V Melocchi for the support with TCGA data mining and analysis, C Mondello for mouse genomic DNA, J Caceres for the T7-tagged SRSF3 expression vector, V Lemmon and PO Couraud for providing the RaL1cd antibody and hCMEC/D3 cells, respectively, and PP Di Fiore for support. We also thank the patients who generously agreed to donate their surgical samples, and the staff of the Division of Gynecology and of the Biobank of European Institute of Oncology for collecting and annotating the human samples. This work was supported by grants from the Associazione Italiana per la Ricerca sul Cancro (AIRC, projects IG-17395 to CG[2], IG-14622 to UC, IG-18683 to ED and IG-18853 to RG, IG-19919 to DG), Worldwide Cancer Research (formerly known as Association for International Cancer Research; AICR 10–0091 to UC), the European Research Council (ERC) under the European Union's Horizon 2020 program (ERC-StG-LS2-637591 to MI) and the Spanish Ministerio de Economía y Competitividad (BFU2017-89201-P to MI). EB, FA and ADM were supported by AIRC - FIRC ITALY postdoctoral fellowships. MG was supported by a fellowship from Fondazione IEO-CCM. Research in FF laboratory is supported by The Giovanni Armenise-Harvard Foundation, and by the Italian Ministry for Education, University and Research (MIUR): Programma Giovani Ricercatori "Rita Levi-Montalcini" and Dipartimenti di Eccellenza Program (2018–2022) - Dept. of Biology and Biotechnology "L. Spallanzani", University of Pavia. AC is supported by a Marie Curie Individual Fellowship from the Horizon 2020 EU Program (Grant agreement n. 745934 – COTETHERS). CG[2] is a consultant for Gene Tools, LLC JS and JM are affiliated with Gene Tools, LLC. Funding bodies had no role in the design of the study and collection, analysis, and interpretation of data and in writing the manuscript.

## Additional information

### Competing interests

Elisabetta Dejana: Reviewing editor, *eLife*. Jon D Moulton, James Summerton: Affiliated with Gene Tools, LLC. Funding bodies had no role in the design of the study and collection, analysis, and interpretation of data and in writing the manuscript. Claudia Ghigna: Consultant for Gene-Tools, LLC. Funding bodies had no role in the design of the study and collection, analysis, and interpretation of data and in writing the manuscript. The other authors declare that no competing interests exist.

## Funding

| Funder | Grant reference number | Author |
| --- | --- | --- |
| Associazione Italiana per la Ricerca sul Cancro | IG-17395 | Claudia Ghigna |
| Worldwide Cancer Research | AICR 10-0091 | Ugo Cavallaro |
| Horizon 2020 Framework Programme | ERC-StG-LS2-637591 | Manuel Irimia |
| Ministerio de Economía y Competitividad | BFU2017-89201-P | Manuel Irimia |
| Giovanni Armenise-Harvard Foundation | | Federico Forneris |
| Ministero dell'Istruzione dell'Università e della Ricerca | | Federico Forneris |
| Associazione Italiana per la Ricerca sul Cancro | IG-14622 | Ugo Cavallaro |
| Associazione Italiana per la Ricerca sul Cancro | IG-18853 | Raffaella Giavazzi |
| Associazione Italiana per la Ricerca sul Cancro | IG-18683 | Elisabetta Dejana |

The funders had no role in study design, data collection and interpretation, or the decision to submit the work for publication.

## Author contributions

Francesca Angiolini, Elisa Belloni, Marco Giordano, Conceptualization, Data curation, Writing—original draft, Writing—review and editing; Matteo Campioni, Federico Forneris, Maria Paola Paronetto, Michela Lupia, Chiara Brandas, Davide Pradella, Anna Di Matteo, Costanza Giampietro, Giovanna Jodice, Chiara Luise, Giovanni Bertalot, Stefano Freddi, Matteo Malinverno, Antonella Chiapparino, Daniel Nyqvist, Davide Gabellini, Data curation; Manuel Irimia, Data curation, Funding acquisition; Jon D Moulton, James Summerton, Resources; Carmen Ghilardi, Elisabetta Dejana, Data curation, Writing—review and editing; Raffaella Giavazzi, Data curation, Funding acquisition, Writing—review and editing; Ugo Cavallaro, Conceptualization, Data curation, Supervision, Funding acquisition, Writing—original draft, Project administration, Writing—review and editing; Claudia Ghigna, Conceptualization, Funding acquisition, Writing—original draft, Project administration, Writing—review and editing

## Author ORCIDs

Elisa Belloni (iD) http://orcid.org/0000-0001-6580-5553
Marco Giordano (iD) http://orcid.org/0000-0001-6335-0030
Federico Forneris (iD) https://orcid.org/0000-0002-7818-1804
Davide Pradella (iD) https://orcid.org/0000-0002-8649-7817
Costanza Giampietro (iD) https://orcid.org/0000-0001-5229-3835
Giovanni Bertalot (iD) https://orcid.org/0000-0002-4862-7705
Jon D Moulton (iD) https://orcid.org/0000-0002-4977-0802
Davide Gabellini (iD) https://orcid.org/0000-0002-3811-4121
Ugo Cavallaro (iD) http://orcid.org/0000-0002-0884-6460
Claudia Ghigna (iD) http://orcid.org/0000-0003-0362-783X

## Ethics

Human subjects: Fresh tissue samples were obtained upon informed consent from patients undergoing surgery at the Gynecology Division of the European Institute of Oncology (Milan). Sample collection was performed under the protocol number R789-IEO approved by the Ethics Committee of the European Institute of Oncology.
Animal experimentation: All animal studies were performed following a protocol approved by the fully authorized animal facility of European Institute of Oncology and by the Italian Ministry of Health

(as required by the Italian Law) (IACUCs number 1256/2015) and in accordance to EU directive 2010/63. Mouse tissues were obtained from Karolinska Institutet (Stockholm, Sweden) and IRCCS San Raffaele Scientific Institute (Milan, Italy), in accordance to Institutional Animal Care and Use Committees.

## Decision letter and Author response
Decision letter https://doi.org/10.7554/eLife.44305.025
Author response https://doi.org/10.7554/eLife.44305.026

# Additional files

## Supplementary files
• Supplementary file 1. Primers used in RT-PCR, RT-qPCR and cloning experiments.
DOI: https://doi.org/10.7554/eLife.44305.017

• Supplementary file 2. Information regarding the re-analysis of AS events affected by Nova2 deple-tion in ECs. The table format corresponds to the default output table from *vast-tools*. Event_ID cor-responds to the AS event identifier used in *vast-tools* and VastDB. Coordinate: coordinates are referred to NCBI37/mm9 genome assembly. Full_coordinate: full set of genomic coordinates of the AS event. AS_Type: Alt3/Alt5, alternative splice site acceptor/donor selection; IR, intron retention; AltEx, cassette alternative exons (including micro-exons when length ≤27 nt). For each sample, two columns provide AS information as obtained from *vast-tools* (PSI, Endoth_Nova2_cont/_KD-PSI and quality scores, Endoth_Nova2_cont/_KD-Q). Further details can be obtained at: https://github.com/vastgroup/vast-tools/blob/master/README.md#combine-output-format.
DOI: https://doi.org/10.7554/eLife.44305.018

• Transparent reporting form
DOI: https://doi.org/10.7554/eLife.44305.019

## Data availability
All data generated or analysed during this study are included in the manuscript and supporting files (Supplementary file 2). The RNA sequencing analysis from The Cancer Genome Atlas (https://portal.gdc.cancer.gov/projects/TCGA-OV) can be downloaded here https://gdc.xenahubs.net/download/TCGA-OV/Xena_Matrices/TCGA-OV.htseq_fpkm.tsv.gz.

The following previously published datasets were used:

| Author(s) | Year | Dataset title | Dataset URL | Database and Identifier |
|---|---|---|---|---|
| Uhlen M, Fager-berg L, Hallstrom BM, Lindskog C, Oksvold P, Mardi-noglu A, Sivertsson A, Kampf C, Sjos-tedt E, Asplund A, Olsson I, Edlund K, Lundberg E, Navani S, Szigyarto CAK, Odeberg J, Djurei-novic D, Takanen JO, Hober S, Alm T | 2015 | Human Protein Atlas project | https://www.proteinatlas.org/ENSG00000104967-NOVA2/pathology/tis-sue/ovarian+cancer#Lo-ocation | The Human Protein Atlas, NOVA2 |
| Giampietro C, De-florian G, Gallo S, Di Matteo A, Pra-della D, Bonomi S, Belloni E | 2015 | NCBI Sequence Read Archive | https://www.ncbi.nlm.nih.gov/bioproject/?term=PRJNA293346 | NCBI BioProject, PRJNA293346 |

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
