## [Decision Letter]

[Editors’ note: a previous version of this study was rejected after peer review, but the authors submitted for reconsideration. The first decision letter after peer review is shown below.]

Thank you for submitting your work entitled "A Novel L1 Isoform with Angiogenic Activity Generated by NOVA2-mediated Alternative Splicing" for consideration by *eLife*. Your article has been reviewed by three peer reviewers, and the evaluation has been overseen by a Reviewing Editor and a Senior Editor. The following individual involved in review of your submission has agreed to reveal their identity: Patrick A Murphy (Reviewer #1).

Based on discussions between the reviewers and editors and the individual reviews below, we regret that we are rejecting your paper for publication in *eLife*.

All the reviewers and editors agreed that the basic results characterizing Nova2 regulation of L1 splicing and its possible role in tumor angiogenesis were quite interesting. However, there was general agreement that several key issues regarding angiogenic effects of the splice variant were not well supported. In particular, it is important to distinguish effects of the soluble splice variant from those of the extracellular domain that is cleaved from the typical isoform. Similarly, it is not clear whether the increased L1 and Nova2 expression observed in tumors was simply due to their increased vascularization and not causal as is proposed. To resolve this, the expression of the L1 variant and Nova2 need better quantification in normal and tumor endothelium. These issues are fundamental to the model that the L1 splice variant is driving tumor angiogenesis. The reviewers raise other valid points below.

*Reviewer #1:*

This is an interesting follow-up on previous work by this group, which had revealed an important function for the Nova2 splice factor in vascular development. Here, they reveal a connection between Nova2 and a target splice event in L1cam, resulting in a form of the protein with altered cellular localization and function. Furthermore, Nova2, which like L1cam is fairly endothelial specific in ovarian tumors, correlates with tumor progression. Overall, the work is well done and exciting, but I think there are a few points which could be addressed in greater detail.

How much of the angiogenic properties of L1cam are due to altered splicing is a critical question. Rather than comparing angiogenic properties of supernatants of cells with exon25 deficient version of L1cam (L1-ΔTM) to vector control, I would have liked to see a comparison between cells expressing exon25 exclusive and inclusive versions of L1cam (L1- ΔTM and L1-FL). This would help address how much of the angiogenic effects observed are due to the change in splicing, rather than L1cam expression alone. It would be best to do these experiments in the matrigel plug assay, which more closely indicates angiogenic function than the tubulogenesis assay (fibroblasts and cancer cells also form tubules on matrigel, PMID: 11806243).

A link between FGF and Exon25- negative L1cam is suggested but not confirmed by the experiments. Is there a change in FGF signaling that can be defined by immunofluorescence or Western blot? For example, could a difference in FGF receptor expression or phosphorylation be detected following treatment with soluble L1cam? One interpretation is that the FGF and L1cam signaling pathways are parallel inputs in the tube formation assay where both are required for tubulogenesis. Given these concerns and the use of the in vitro tube formation assay alone as an indication of their cross-talk, the claim that "L1-ΔTM promotes angiogenesis via FGFR1" seems overstated.

Authors allude to data on increased secretion of L1cam in endothelial cells (Discussion, first paragraph), but I do not see a data reference for this observation. Can they measure soluble L1cam with the C-terminus in the supernatant of their endothelial cell lines and some other L1 expressing cells which do not make the L1-ΔTM isoform? Or can they measure this in their endothelial cells with and without Nova2 suppression? This would help alleviate concerns about any artifacts due to overexpression of L1cam in moECs.

If L1-ΔTM is specific to tumor endothelium, it is not surprising that tumor levels are correlated with vascular density. It would be more meaningful if the endothelial levels of L1-ΔTM expression were correlated with tumor vascularization. There is some heterogeneity in the L1-ΔTM expression levels in isolated tumor endothelial cells (Figure 5C), do those with lower L1-ΔTM come from less vascular tumors? Do they have lower NOVA2 expression? Perhaps that is what the authors looked at already, but this was not clear to me.

*Reviewer #2:*

L1 (L1CAM gene) is a cell-surface glycoprotein that undergoes ectodomain shedding via proteolytic cleavage at a membrane proximal site. This soluble ectodomain has previously been implicated in a wide array of cell signaling and adhesion processes involved in tumorigenesis.

In this manuscript, Angiolini et al. identify a novel isoform of L1CAM with exon 25 skipped, termed L1-ΔTM. Because exon 25 contains the entire transmembrane domain of the protein, skipping of this exon results in a L1 protein isoform that no longer localizes on the cell surface and instead is found both in the cytoplasm and secreted into the extracellular space. The authors investigate whether this secreted full-length protein isoform of L1 (which does not undergo proteolytic cleavage and maintains the cytoplasmic tail) drives a pro-angiogenic program that supports aggressive tumorigenesis. They identify the well-characterized splicing factor, Nova2, as an important factor in promoting skipping of exon 25.

This manuscript presents a novel alternative (non-proteolytic) mechanism for the release of the L1 ectodomain into the extracellular space and characterizes the role of Nova2 in producing this isoform. The observation that this novel splice variant occurs in endothelial cells and its regulation by Nova2, as well as the exogenous expression of the full-length and L1-ΔTM isoforms to produce glycosylated transmembrane and soluble proteins, respectively, are all compelling pieces of data. However, the major flaw in the manuscript is the lack of physiological evidence that the angiogenic effects observed upon overexpression of L1-ΔTM are specific to the isoform itself, or whether a similar pro-angiogenic affect would also result from overexpressing full length L1. Indeed, overexpression of membrane bound L1 (FL-L1) resulted in release of L1 into the extracellular space (Figure 2D), presumably through proteolytic cleavage which would presumably have the same effect.

The difficulty in separating the effect of the cleaved full-length protein compared to the exon 25 skipped splice variant is acknowledged. The authors could address this issue by generating a mutation in the proteolytic cleavage site that would render it inactive and determine whether the splicing variant can overcome this deficiency.

A separate concern related to this criticism is that the manuscript relies almost entirely on a gain-of-function system. It is unclear how the level of overexpression of the L1-ΔTM isoform relates to endogenous expression either in cell lines or tumors, and without the L1-FL controls, one cannot assess the relative potency of this isoform being expressed.

*Reviewer #3:*

The manuscript by Angiolini et al. describes a novel alternatively spliced isoform of the cell-surface adhesion molecule L1CAM in endothelial cells (ECs). This novel isoform lacking exon 25, lacks the transmembrane domain, creating a soluble form of L1CAM. The authors show that this soluble L1-ΔTM isoform enhances the formation of tubes in mouse endothelial cells and this is inhibited by an inhibitor of FGFR1 signaling. They show direct binding of NOVA2 to exon 25 of the L1CAM transcript and in a splicing assay, show NOVA2 to be responsible for promoting skipping of L1CAM exon 25. Finally, they show a correlation between higher vessel density in L1-ΔTM-positive ovarian tumors, implying that alternative splicing of L1CAM correlates with tumor vascularization, supporting a proangiogenic function for L1-ΔTM.

In general this manuscript is well written and easy to follow. The experimental data is clearly presented. The novelty of this report is the identification of a cell adhesion molecule converted into a soluble angiogenic factor through alternative splicing. This represents not only a novel target for future antiangiogenic strategies but also can be used for diagnostic and prognostic purposes. However, there are some things lacking in the paper that need to be addressed.

1) Even though NOVA2 is an important splicing factor in EC cells, there is a bit of a logic "jump" caused by focusing only on NOVA2 as the regulator of L1CAM. The authors need to show other possible splicing factors from the SFmap/SpliceAid analysis, and possibly knockdown several of them (e.g. SRSF1, SRSF2, PTBP1, in addition to hnRNP A1) in their minigene splicing assay or RT-PCR on the endogenous transcripts.

2) Figure 5 The authors need to normalize the amount of L1CAM and NOVA2 to the amount of blood vessels in the IHC. If any blood vessel expresses L1CAM and NOVA2 then it is only a measure of higher vascularity in the tumor. The relevant question is if the blood vessels in the tumor express higher L1CAM and NOVA2 than normal? Also, it seems that additional cells are stained with NOVA2 (epithelial cells or other cells?) which do not express L1CAM.

3) In order to claim that the blood vessels in the tumor express more NOVA2 and L1CAM isoform, the authors need to show that in the presence of the cancer cells the EC cells express higher levels of NOVA2. Maybe this is through secretion of factors from the cancer cells or direct interaction, but the claim is not proven until some causative experiment is performed.

4) Figure 4—figure supplement 3B and C should show endogenous levels of Nova2 in the transfected cells. The authors write in Materials and methods "Since NOVA2 expression is regulated by EC density (Giampietro et al., 2015), for the analysis of L1 splicing, NOVA2-knockdown moEC were used as confluent monolayers (500,000 cells in 35-mm Petri dishes), whereas moEC overexpression HA-tagged NOVA2 were tested at low density (500,000 cells in 100-mm Petri dishes)." If the expression is sensitive to cell density it should be shown for the specific experiments.

5) Figure 4—figure supplement 3F. The effect of Nova2 overexpression on L1CAM splicing in human cells (82% inclusion) is not as strong as in mouse cells Figure 4B (27.2% inclusion). Can the authors provide an explanation for that?

[Editors’ note: what now follows is the decision letter after the authors submitted for further consideration.]

Thank you for submitting your article "A Novel L1 Isoform with Angiogenic Activity Generated by NOVA2-mediated Alternative Splicing" for consideration by *eLife*. Your article has been reviewed by three peer reviewers, and the evaluation has been overseen by Douglas Black as Reviewing Editor and Jonathan Cooper as the Senior Editor. The following individual involved in review of your submission has agreed to reveal their identity: Patrick A Murphy (Reviewer #1).

The reviewers have discussed the reviews with one another and the Reviewing Editor has drafted this decision to help you prepare a revised submission.

This manuscript is a resubmission of a previously rejected study. The reviewers agreed that the previous version was of significant interest but all had major criticisms and experiments that they felt were needed to support its conclusions. All of the reviewers now agree that the authors have made major changes that strengthen the paper. Assuming that the authors can adequately address the remaining points in these reviews, the editors are pleased to accept this paper for publication in *eLife*.

*Reviewer #1:*

The addition of new data has solidified key conclusions and addressed my original concerns. With these new pieces of data, the authors make a compelling argument that a reduction in the inclusion of exon 25, which is mediated by Nova2 in the endothelium, promotes the expression of a secreted L1 variant with angiogenic properties beyond the constitutive form of L1. I particularly like the addition of Supplementary Figure 2M and N, showing that a morpholino suppressing L1 exon 25 inclusion results in increased secretion of soluble L1. I think this should be considered for the main figures.

*Reviewer #2:*

This manuscript is a resubmission of a previously reviewed work that all reviewers agreed presents an important and novel biological mechanism in which alternative splicing essentially replaces proteolytic cleavage to release a polypeptide that is otherwise integrally associated with the plasma membrane. The authors have addressed most of the initial reviewers' comments and the paper is clearly much stronger now from the addition of new data and textual clarifications.

I had originally suggested a making a mutation in the proteolytic site – Figure 2A showing the putative ADAM cleavage site as a box implies some knowledge of where cleavage occurs (hence the suggested experiment, as there are only 6 amino acids in the extracellular domain of that region). If it is not clear where the cleavage occurs (and which enzymes are responsible), this figure is a bit misleading. The difficulties the authors cite regarding performing the suggested experiments are reasonable, but perhaps the figure could be clarified.

The immunofluorescence data added to Figure 2C is much improved with regard to clearly showing the subcellular localization of each isoform. The data that has been added to Figure 3 (and in particular Figure 3F) are instrumental in addressing both my concerns and those of reviewer #1, this experiment adds a great deal of support to the central hypothesis of this paper.

The authors have taken the criticism that the previous manuscript was overly dependent on gain-of-function experiments constructively in adding the new data presented in Figure 2—figure supplement 1. These experiments provide an elegant alternative approach to overexpression of individual isoforms by perturbing the splicing of the endogenous gene. This is a well-conceived and executed experiment.

Figure 5B solidifies the assertion that NOVA2 and L1 are colocalized. This is a strong piece of evidence for the physiological relevance of the proposed mechanism within tumors.

In summary, with this resubmission the authors have more than satisfactorily addressed the major points of concern raised in the original manuscript.

*Reviewer #3:*

The manuscript "A novel L1 isoform with angiogenic activity generated by Nova-2 mediated alternative splicing" has been revised. The authors have supplied additional data in several figures, new supplementary figures and table and two figures in their response to the reviewers. In addition, they have addressed the concerns of the reviewers in their letter.

They have analyzed a cohort of 13 ovarian cancer samples for vessel density and expression levels of L1-ΔTM. They show that the levels of mRNA correlate with the degree of tumor vascularization (Figure 5D). They have also redone the immunohistochemistry of NOVA2 in EC cells (Figure 5B) showing that the immunoreactivity of NOVA2 is restricted to the nuclei of vascular ECs.

The authors did not completely address point 3. Thus, the connection between L1-ΔTM and vascularization is left at the level of correlation rather than a causative effect. The authors should at least say that further experiments are required to determine if this a direct effect of L1-ΔTM.

---

## [Author Response]

[Editors’ note: the author responses to the first round of peer review follow.]

Reviewer #1:This is an interesting follow-up on previous work by this group, which had revealed an important function for the Nova2 splice factor in vascular development. Here, they reveal a connection between Nova2 and a target splice event in L1cam, resulting in a form of the protein with altered cellular localization and function. Furthermore, Nova2, which like L1cam is fairly endothelial specific in ovarian tumors, correlates with tumor progression. Overall, the work is well done and exciting, but I think there are a few points which could be addressed in greater detail.How much of the angiogenic properties of L1cam are due to altered splicing is a critical question. Rather than comparing angiogenic properties of supernatants of cells with exon25 deficient version of L1cam (L1-ΔTM) to vector control, I would have liked to see a comparison between cells expressing exon25 exclusive and inclusive versions of L1cam (L1-ΔTM and L1-FL). This would help address how much of the angiogenic effects observed are due to the change in splicing, rather than L1cam expression alone. It would be best to do these experiments in the matrigel plug assay, which more closely indicates angiogenic function than the tubulogenesis assay (fibroblasts and cancer cells also form tubules on matrigel, PMID: 11806243).

We have performed the matrigel plug assay as recommended by the reviewer. As shown in Figure 3F, and described in the last paragraph of the subsection “L1-ΔTM regulates endothelial cell function”, the conditioned medium (CM) from L1-ΔTM- expressing ECs was significantly more angiogenic than the CM from L1-FL-expressing cells. Incidentally, we obtained similar results also in the tubulogenesis assay (see Author response image 1). Based on the reviewer’s concern about the tube formation assay, we did not include these in vitrodata in the revised manuscript, and just mentioned them as data not shown in the second paragraph of the aforementioned subsection.

These additional experiments indicated that L1-ΔTM per seis endowed with angiogenic properties, while the mere expression of L1cam has little effect, further supporting the biological relevance of the splicing of exon 25 in endothelial cells.

**Author response image 1. respfig1:** L1-ΔTM promotes endothelial tube formation. Parental moEC were subjected to tube formation assays in the presence of the CM from moEC transduced with either the empty vector, L1-FL or L1- ΔTM. The CM from L1- ΔTM-expressing cells exhibited higher in vitroangiogenic potential than that from L1-FL- expressing cells or from control cells. ***p*<0.01; **p*<0.05.

A link between FGF and Exon25- negative L1cam is suggested but not confirmed by the experiments. Is there a change in FGF signaling that can be defined by immunofluorescence or Western blot? For example, could a difference in FGF receptor expression or phosphorylation be detected following treatment with soluble L1cam? One interpretation is that the FGF and L1cam signaling pathways are parallel inputs in the tube formation assay where both are required for tubulogenesis. Given these concerns and the use of the in vitro tube formation assay alone as an indication of their cross-talk, the claim that "L1-ΔTM promotes angiogenesis via FGFR1" seems overstated.

As suggested by the reviewer, we investigated whether soluble L1-ΔTM affects FGFR phosphorylation. Treating parental moEC with recombinant L1-ΔTM resulted in increased phospho-FGFR1, consistent with the L1-ΔTM-induced activation of FGFR1. These observations have been included in the Results (Figure 3G and subsection “FGFR1 signaling is required for L1-ΔTM-induced tube formation”).

The reviewer is correct in pointing out that FGFR1 and L1CAM might activate parallel pathways that are both involved in tube formation. However, the observation that the FGFR inhibitor PD173074 reduces the tubulogenesis of L1-ΔTM-expressing moEC to the level of control cells (but has no effect on the L1-ΔTM-independent tube formation in control moEC), strongly suggests that in the context of tubulogenesis FGFR1 signaling acts as an effector of L1-ΔTM. Nevertheless, we agree with the reviewer’s concern about the overstatement and have modified the sentence accordingly (“FGFR1 signaling is required for L1-ΔTM- induced tube formation”).

Authors allude to data on increased secretion of L1cam in endothelial cells (Discussion, first paragraph), but I do not see a data reference for this observation. Can they measure soluble L1cam with the C-terminus in the supernatant of their endothelial cell lines and some other L1 expressing cells which do not make the L1-ΔTM isoform? Or can they measure this in their endothelial cells with and without Nova2 suppression? This would help alleviate concerns about any artifacts due to overexpression of L1cam in moECs.

With regard to the concern about artifacts, we would argue that the data shown in Figure 2—figure supplement 1 reduce significantly the risk of artifacts due to the overexpression of L1CAM. The figure, indeed, shows in both moEC and luEC that the ectopic expression of L1-FL per se (shown in Figure 2—figure supplement 1A, lower panel, and Figure 2—figure supplement 1B, right upper panel) does not result in the secretion of significant amounts of L1 with the C-terminus. The latter is detected occurs only in the CM of L1-ΔTM- expressing cells (Figure 2—figure supplement 1A, upper panel, and Figure 2—figure supplement 1B, right lower panel), supporting the specificity of our results.

Nevertheless, the reviewer’s suggestion to compare endothelial cells with another L1- expressing cell type was very insightful. We compared mouse lung-derived EC, which express endogenous L1-ΔTM (Figure 1C, “lu2EC”) and the mouse melanoma cell line B16, known to express L1 (Linnemann and Bock, 1986; Magrini et al., 2014). We first verified by RT-PCR that B16 cells do not express the L1-ΔTM isoform (Figure 2—figure supplement 1E), which is consistent with the lack of Nova2 expression (Figure 2—figure supplement 1E), and then assayed for the secretion of C- terminus-containing L1 in the conditioned medium. Despite B16 cells express much higher levels of endogenous, cell-associated L1 than lu2EC (Figure 2—figure supplement 1F), no protein was detectable in their conditioned medium with the antibody against the cytoplasmic tail, while lu2EC secreted high amounts of C-terminus-containing L1 (Figure 2—figure supplement 1F). Together with the other data presented in the manuscript, these findings (described in the fourth paragraph of the subsection “Alternative splicing generates a novel soluble form of L1”) strongly support the increased secretion of L1- ΔTM in endothelial cells.

If L1-ΔTM is specific to tumor endothelium, it is not surprising that tumor levels are correlated with vascular density. It would be more meaningful if the endothelial levels of L1-ΔTM expression were correlated with tumor vascularization. There is some heterogeneity in the L1-ΔTM expression levels in isolated tumor endothelial cells (Figure 5C), do those with lower L1-ΔTM come from less vascular tumors? Do they have lower NOVA2 expression? Perhaps that is what the authors looked at already, but this was not clear to me.

We agree with the reviewer that “it would be more meaningful if the endothelial levels of L1-ΔTM expression were correlated with tumor vascularization”. Unfortunately, we could not assess the vascularization of the same tumors used to derive endothelial cells, as proposed by the reviewer, since the entire surgical specimens were processed to obtain the ECs themselves. Nevertheless, since our data strongly suggest that tissue expression of L1-ΔTM is largely confined to vascular endothelium (e.g., Figure 1B, C and Figure 5B), we performed RT-PCR for L1-ΔTM on a cohort of 17 archival samples of ovarian cancer. The same specimens were also stained for CD31 to measure vessel density. As shown in Figure 5D, the levels of L1-ΔTM mRNA correlated with the degree of tumor vascularization (*r*=0.84, p<0.001).

Reviewer #2:[…] This manuscript presents a novel alternative (non-proteolytic) mechanism for the release of the L1 ectodomain into the extracellular space and characterizes the role of Nova2 in producing this isoform. The observation that this novel splice variant occurs in endothelial cells and its regulation by Nova2, as well as the exogenous expression of the full-length and L1-ΔTM isoforms to produce glycosylated transmembrane and soluble proteins, respectively, are all compelling pieces of data. However, the major flaw in the manuscript is the lack of physiological evidence that the angiogenic effects observed upon overexpression of L1-ΔTM are specific to the isoform itself, or whether a similar pro-angiogenic affect would also result from overexpressing full length L1. Indeed, overexpression of membrane bound L1 (FL-L1) resulted in release of L1 into the extracellular space (Figure 2D), presumably through proteolytic cleavage which would presumably have the same effect.

We thank reviewer #2 for her/his positive comments and for the constructive criticisms that allowed us to improve our manuscript.

With regard to the L1-ΔTM-specific effect, as specified also in our reply to reviewer #1, we have performed both in vivo matrigel plug and in vitro tube formation assays to compare cells expressing L1-ΔTM vs. full-length L1. As shown in Figure 3F and described in the last paragraph of the subsection “L1-ΔTM regulates endothelial cell function”, the conditioned medium (CM) from L1-ΔTM-expressing ECs was significantly more angiogenic than the CM from L1-FL-expressing cells. Similar results were obtained in the tubulogenesis assay (see Author response image 1). As recommended by reviewer #1, only the results of the Matrigel plug assay have been included in the manuscript.

The difficulty in separating the effect of the cleaved full-length protein compared to the exon 25 skipped splice variant is acknowledged. The authors could address this issue by generating a mutation in the proteolytic cleavage site that would render it inactive and determine whether the splicing variant can overcome this deficiency.

We agree with the reviewer that this would be a very elegant and appropriate strategy. However, it would be quite a challenging task, given that the proteolytic cleavage of L1 can be carried out by a plethora of proteases (ADAM10, ADAM17, plasmin, trypsin, proprotein convertase PC5A, etc.). Moreover, L1 shedding has been investigated mostly in neurons, and the enzymes involved in the cleavage of L1 in endothelial cells remain elusive. Therefore, it would be quite difficult and time-consuming to identify such enzymes and perform the mutagenesis suggested by reviewer #2, and we feel that such an effort would fall out of the scope of our manuscript.

A separate concern related to this criticism is that the manuscript relies almost entirely on a gain-of-function system. It is unclear how the level of overexpression of the L1-ΔTM isoform relates to endogenous expression either in cell lines or tumors, and without the L1-FL controls, one cannot assess the relative potency of this isoform being expressed.

We thank the reviewer for asking these important controls, which lent further support to our conclusions. As described above in our response to reviewer #1, we have included the requested L1-FL controls. The results showed that L1-ΔTM-containing conditioned medium (CM) has a markedly higher angiogenic potential than L1-FL CM (Figure 3F and Author response image 1).

To address the concern on the manuscript relying almost entirely on a gain-of-function system, we have added new data on an endogenous system. The lung-derived endothelial cells lu2EC, which express both NOVA2 and L1-ΔTM endogenously (Figures 1C and Figure 2—figure supplement 1E), were treated with a morpholino oligonucleotide that selectively interfered with the inclusion of exon 25 (Figure 3E). As shown in Figure 3F (left), this resulted in increased skipping of exon 25 and release of endogenous L1-ΔTM in the extracellular compartment. Importantly, ECs exposed to the CM from morpholino-treated cells exhibited higher tube-forming activity than those exposed to control CM (Figure 3F, right). These observations validated in an endogenous system our observations on the angiogenic activity of AS-generated L1-ΔTM.

Reviewer #3:[…] 1) Even though NOVA2 is an important splicing factor in EC cells, there is a bit of a logic "jump" caused by focusing only on NOVA2 as the regulator of L1CAM. The authors need to show other possible splicing factors from the SFmap/SpliceAid analysis, and possibly knockdown several of them (e.g. SRSF1, SRSF2, PTBP1, in addition to hnRNP A1) in their minigene splicing assay or RT-PCR on the endogenous transcripts.

As outlined in more details in the Results (paragraph “NOVA2 controls alternative splicing of L1-ΔTM in ECs”), the SFmap analysis identified NOVA2, hnRNP A1 and SRSF3 as possible candidate regulators of *L1cam* exon 25 splicing (Figure 4—figure supplement 1A). By using the minigene- based splicing assay as suggested by the reviewer, we found that the skipping of *L1cam* exon 25 occurred only upon overexpression of NOVA2, but not with hnRNP A1 or SRSF3 (Figure 4B and Figure 4—figure supplement 1B).

To further probe the specificity of NOVA2, we also tested SRSF1, as an additional AS regulator, and two known splicing repressors, hnRNP M and hnRNP A2/B1. None of these additional candidates was able to affect skipping of *L1cam* exon 25 (data not included in the revised manuscript but shown in Author response image 2).

The rationale for investigating NOVA2-mediated AS of *L1cam* was also supported by the analysis of the RNA-seq data from NOVA2-knockdown ECs (Giampietro et al., 2015), which revealed *L1cam* exon 25 as a novel target of NOVA2 in ECs (subsection “NOVA2 controls alternative splicing of L1-ΔTM in ECs”, second paragraph, and Supplementary file 2).

**Author response image 2. respfig2:** Evaluation of candidate splicing regulatory factors (SRFs) on L1cam splicing. AS of the p-L1 wild- type WT minigene co-transfected in HeLa cells with either HA-NOVA2, FLAG-hnRNP A2B1 (FLAG-A2B1), GFP- hnRNP M (GFP-M) or the empty vector. Top panels show that, in contrast to NOVA2, none of the other SRFs affected the splicing of L1cam exon 25. Bottom panels show the ectopic expression of each SRF as revealed by western blotting with the indicated antibodies (α-Tubulin as loading control).

2) Figure 5 The authors need to normalize the amount of L1CAM and NOVA2 to the amount of blood vessels in the IHC. If any blood vessel expresses L1CAM and NOVA2 then it is only a measure of higher vascularity in the tumor. The relevant question is if the blood vessels in the tumor express higher L1CAM and NOVA2 than normal? Also, it seems that additional cells are stained with NOVA2 (epithelial cells or other cells?) which do not express L1CAM.

Figure 5A (left panel) shows indeed the percentage of NOVA2-positive vessels over the total number of vessels (determined by CD31 staining). We have now clarified this in the legend to Figure 5A as well as on the Y-axis of the graph. In addition, we now provide analogous measurements also for L1-positive vessels, confirming that they are markedly increased in cancer samples (Figure 5A, right panel).

Reviewer #3 argues that “if any blood vessel expresses L1CAM and NOVA2 then it is only a measure of higher vascularity in the tumor”. This concern was somehow similar to the last major comment of reviewer #1. To investigate in more detail the possible correlation between the expression of L1-ΔTM and tumor angiogenesis, we have analysed a cohort of 13 ovarian cancer samples both for vessel density and for the expression levels of L1-ΔTM (for more details, see our reply to the last major comment of reviewer #1). As shown in Figure 5D, the levels of L1-ΔTM mRNA correlated with the degree of tumor vascularization (*r*=0.84, p<0.001).

As for the expression of NOVA2 in additional cell types, we acknowledge that from the original IHC images one could hypothesize the presence of NOVA2 in non-endothelial cells. We have investigated that issue in more detail and found out that the faint cytosolic staining was due to non-specific interactions of the antibodies. Indeed, we have now optimized the immunohistochemistry conditions (using Viva Green instead of diaminobenzidine as the chromogenic substrate) and are able to show that the immunoreactivity of NOVA2 is restricted to the nuclei of vascular ECs (Figure 5B and Figure 5—figure supplement 2A). Finally, the vessel-specific immunoreactivity of NOVA2 is also shown in the images taken from the Human Protein Atlas website (Figure 5—figure supplement 1B).

3) In order to claim that the blood vessels in the tumor express more NOVA2 and L1CAM isoform, the authors need to show that in the presence of the cancer cells the EC cells express higher levels of NOVA2. Maybe this is through secretion of factors from the cancer cells or direct interaction, but the claim is not proven until some causative experiment is performed.

We are afraid we did not understand completely the reviewer’s comment. In particular, we do not see the causal link between the effect of cancer cells on the endothelial expression of NOVA2 and our “claim that the blood vessels in the tumor express more NOVA2 and L1CAM isoform”. Along the same line, it is unclear to us why “the claim is not proven until some causative experiment [on the role of cancer cells] is performed”. The increased expression of NOVA2 and L1CAM in ovarian cancer vasculature is illustrated in Figure 5A, where we show that tumors contain 6 and 4-fold more vessels positive for NOVA2 and L1CAM, respectively, than their normal tissue counterpart. This phenomenon is independent on whether and how cancer cells regulate the expression of the two proteins in the endothelium.

Having said that, we agree with the reviewer about the relevance of understanding the mechanisms that regulate the expression of NOVA2 in tumor-associated ECs. Nevertheless, we feel that investigating this issue would fall out of the scope of the present study. The experiments suggested by the reviewer, in addition, would be quite time-consuming in that they should not be limited to the cancer cells themselves, but should rather be extended to other components of the tumor microenvironment. It is conceivable, indeed, that different cell types and/or other microenvironmental factors account for (or at least contribute to) the induction of NOVA2 in the cancer vasculature. Along this line, a very recent paper reported that NOVA2 is upregulated in colorectal carcinoma vasculature and that endothelial NOVA2 expression is induced upon exposure to hypoxia (Gallo et al., 2018), a condition shared by many solid tumors. This paper is now cited in our Discussion (sixth paragraph).

4) Figure 4—figure supplement 3B and C should show endogenous levels of Nova2 in the transfected cells. The authors write in Materials and methods "Since NOVA2 expression is regulated by EC density (Giampietro et al., 2015), for the analysis of L1 splicing, NOVA2-knockdown moEC were used as confluent monolayers (500,000 cells in 35-mm Petri dishes), whereas moEC overexpression HA-tagged NOVA2 were tested at low density (500,000 cells in 100-mm Petri dishes)." If the expression is sensitive to cell density it should be shown for the specific experiments.

Figure 4—figure supplement 3A now shows the endogenous levels of *Nova2* and *L1-ΔTM* in moEC cultured as sparse cells or at confluency. Confluent cells exhibit higher *Nova2* expression as compared to sparse cells, at both mRNA and protein levels, thus demonstrating that *Nova2* expression in moEC is sensitive to cell density, in agreement with previous data (Giampietro et al., 2015). The higher level of *Nova2* in confluent moEC is mirrored by an increase in the exclusion of *L1cam* exon 25 (Figure 4—figure supplement 3A).

5) Figure 4—figure supplement 3F. The effect of Nova2 overexpression on L1CAM splicing in human cells (82% inclusion) is not as strong as in mouse cells Figure 4B (27.2% inclusion). Can the authors provide an explanation for that?

While it is possible that the discrepancy involves, at least in part, some species-specific features, also the origin of ECs may play a role in the relative efficiency of NOVA2 in *L1CAM* splicing. In this specific case, moEC derive from mouse lung, while hCMEC/D3 derive from human brain vessels. The heterogeneity of ECs from different districts, in fact, has been extensively documented and may reflect profoundly different phenotypic and functional properties (Potente and Mäkinen, Nat Rev Mol Cell Biol, 2017).

To further address the reviewer’s comment, we have reduced by shRNA the expression of endogenous NOVA2 in a different EC model, consisting of human umbilical vein ECs. As shown in Figure 4—figure supplement 3G, *NOVA2* knockdown resulted in a marked decrease in L1CAM splicing (with the inclusion of exon 25 going from 72% to 100%). This supports the hypothesis that the splicing of exon 25 in response to the genetic manipulation of NOVA2 is quantitatively different indifferent EC types. Of note, these data suggest that the activity of NOVA2 is cell context- dependent even within the same cell type (i.e., endothelial cells).

[Editors' note: the author responses to the re-review follow.]

Reviewer #1:The addition of new data has solidified key conclusions and addressed my original concerns. With these new pieces of data, the authors make a compelling argument that a reduction in the inclusion of exon 25, which is mediated by Nova2 in the endothelium, promotes the expression of a secreted L1 variant with angiogenic properties beyond the constitutive form of L1. I particularly like the addition of Supplementary Figure 2M and N, showing that a morpholino suppressing L1 exon 25 inclusion results in increased secretion of soluble L1. I think this should be considered for the main figures.

We thank reviewer #1 for his appreciation of the work we have done to improve our manuscript. We also thank him for his comments on the data presented in Supplementary Figure 2M and N. Following his suggestion, we have modified Figure 3 to include the morpholino data (Figure 3E and F). In order to avoid overcrowding of the figure, the microscope pictures of the tubes previously shown in Figure 3A have been moved to Figure 3—figure supplement 1A.

Reviewer #2:[…] I had originally suggested a making a mutation in the proteolytic site – Figure 2A showing the putative ADAM cleavage site as a box implies some knowledge of where cleavage occurs (hence the suggested experiment, as there are only 6 amino acids in the extracellular domain of that region). If it is not clear where the cleavage occurs (and which enzymes are responsible), this figure is a bit misleading. The difficulties the authors cite in the letter regarding performing the suggested experiments are reasonable, but perhaps the figure could be clarified.

We agree with the reviewer that Figure 2A was misleading. Therefore, following the reviewer’s suggestion to clarify the figure and considering that our manuscript does not deal with the proteolytic cleavage of L1CAM, we have removed the box indicating the putative ADAM cleavage site.

Reviewer #3:[…] The authors did not completely address point 3. Thus, the connection between L1-ΔTM and vascularization is left at the level of correlation rather than a causative effect. The authors should at least say that further experiments are required to determine if this a direct effect of L1-ΔTM.

We appreciate the reviewer’s thorough analysis of the new data that have been added in the revised manuscript. With regard to the last point, we have modified the Discussion to point out that further work should test the causal role of vascular L1-ΔTM in tumor vascularization (Discussion, sixth paragraph).